# Near-Optimal Time and Sample Complexities for Solving Markov Decision Processes with a Generative Model

**Aaron Sidford**
Stanford University
sidford@stanford.edu

**Mengdi Wang**
Princeton University
mengdiw@princeton.edu

**Xian Wu**
Stanford University
xwu20@stanford.edu

**Lin F. Yang**
Princeton University
lin.yang@princeton.edu

**Yinyu Ye**
Stanford University
yyye@stanford.edu

## Abstract

In this paper we consider the problem of computing an $\epsilon$-optimal policy of a discounted Markov Decision Process (DMDP) provided we can only access its transition function through a generative sampling model that given any state-action pair samples from the transition function in $O(1)$ time. Given such a DMDP with states $\mathcal{S}$, actions $\mathcal{A}$, discount factor $\gamma \in (0, 1)$, and rewards in range $[0, 1]$ we provide an algorithm which computes an $\epsilon$-optimal policy with probability $1 - \delta$ where *both* the time spent and number of sample taken are upper bounded by

$$O\left[ \frac{|\mathcal{S}||\mathcal{A}|}{(1-\gamma)^3 \epsilon^2} \log\left( \frac{|\mathcal{S}||\mathcal{A}|}{(1-\gamma)\delta\epsilon} \right) \log\left( \frac{1}{(1-\gamma)\epsilon} \right) \right] \ .$$

For fixed values of $\epsilon$, this improves upon the previous best known bounds by a factor of $(1-\gamma)^{-1}$ and matches the sample complexity lower bounds proved in [AMK13] up to logarithmic factors. We also extend our method to computing $\epsilon$-optimal policies for finite-horizon MDP with a generative model and provide a nearly matching sample complexity lower bound.

## 1  Introduction

Markov decision processes (MDPs) are a fundamental mathematical abstraction used to model sequential decision making under uncertainty and are a basic model of discrete-time stochastic control and reinforcement learning (RL). Particularly central to RL is the case of computing or learning an approximately optimal policy when the MDP itself is not fully known beforehand. One of the simplest such settings is when the states, rewards, and actions are all known but the transition between states when an action is taken is probabilistic, unknown, and can only be sampled from.

Computing an approximately optimal policy with high probability in this case is known as PAC RL with a generative model. It is a well studied problem with multiple existing results providing algorithms with improved the sample complexity (number of sample transitions taken) and running time (the total time of the algorithm) under various MDP reward structures, e.g. discounted infinite-horizon, finite-horizon, etc. (See Section 5 for a detailed review of the literature.)

In this work, we consider this well studied problem of computing approximately optimal policies of discounted infinite-horizon Markov Decision Processes (DMDP) under the assumption we can only access the DMDP by sampling state transitions. Formally, we suppose that we have a DMDP

with a known set of states, $\mathcal{S}$, a known set of actions that can be taken at each states, $\mathcal{A}$, a known reward $\boldsymbol{r}_{s,a} \in [0, 1]$ for taking action $a \in \mathcal{A}$ at state $s \in \mathcal{S}$, and a discount factor $\gamma \in (0, 1)$. We assume that taking action $a$ at state $s$ probabilistically transitions an agent to a new state based on a fixed, but unknown probability vector $\boldsymbol{P}_{s,a}$. The objective is to maximize the cumulative sum of discounted rewards in expectation. Throughout this paper, we assume that we have a *generative model*, a notion introduced by [Kak03], which allows us to draw random state transitions of the DMDP. In particular, we assume that we can sample from the distribution defined by $\boldsymbol{P}_{s,a}$ for all $(s, a) \in \mathcal{S} \times \mathcal{A}$ in $O(1)$ time. This is a natural assumption and can be achieved in expectation in certain computational models with linear time preprocessing of the DMDP.[1]

The main result of this paper is that we provide the first algorithm that is sample-optimal and runtime-optimal (up to polylogarithmic factors) for computing an $\epsilon$-optimal policy of a DMDP with a generative model (in the regime of $\epsilon \geq 1/\sqrt{(1 - \gamma)|\mathcal{S}|}$). In particular, we develop a randomized Variance-Reduced Q-Value Iteration (vQVI) based algorithm that computes an $\epsilon$-optimal policy with probability $1 - \delta$ with a number of samples, i.e. queries to the generative model, bound by

$$O\left[ \frac{|\mathcal{S}||\mathcal{A}|}{(1 - \gamma)^3 \epsilon^2} \log\left(\frac{|\mathcal{S}||\mathcal{A}|}{(1 - \gamma)\delta\epsilon}\right) \log\left(\frac{1}{(1 - \gamma)\epsilon}\right) \right] .$$

This result matches (up to polylogarithmic factors) the following sample complexity lower bound established in [AMK13] for finding $\epsilon$-optimal policies with probability $1 - \delta$ (see Appendix D):

$$\Omega\left[ \frac{|\mathcal{S}||\mathcal{A}|}{(1 - \gamma)^3 \epsilon^2} \log\left(\frac{|\mathcal{S}||\mathcal{A}|}{\delta}\right) \right] .$$

Furthermore, we show that the algorithm can be implemented using sparse updates such that the overall run-time complexity is equal to its sample complexity up to constant factors, as long as each sample transition can be generated in $O(1)$ time. Consequently, up to logarithmic factors our run time complexity is optimal as well. In addition, the algorithm's space complexity is $\Theta(|\mathcal{S}||\mathcal{A}|)$.

Our method and analysis builds upon a number of prior works. (See Section 5 for an in-depth comparison.) The paper [AMK13] provided the first algorithm that achieves the optimal sample complexity for finding $\epsilon$-optimal value functions (rather than $\epsilon$-optimal policy), as well as the matching lower bound. Unfortunately an $\epsilon$-optimal value function does not imply an $\epsilon$-optimal policy and if we directly use the method of [AMK13] to get an $\epsilon$-optimal policy for constant $\epsilon$, the best known sample complexity is $\widetilde{O}(|\mathcal{S}||\mathcal{A}|(1 - \gamma)^{-5}\epsilon^{-2})$. [2] This bound is known to be improvable through related work of [SWWY18] which provides a method for computing an $\epsilon$-optimal policy using $\widetilde{O}(|\mathcal{S}||\mathcal{A}|(1 - \gamma)^{-4}\epsilon^{-2})$ samples and total runtime and the work of [AMK13] which in the regime of small approximation error, i.e. where $\epsilon = O((1 - \gamma)^{-1/2}|\mathcal{S}|^{-1/2})$, already provides a method that achieves the optimal sample complexity. However, when the approximation error takes fixed values, e.g. $\epsilon \geq \Omega((1 - \gamma)^{-1/2}|\mathcal{S}|^{-1/2})$, there remains a gap between the best known runtime and sample complexity for computing an $\epsilon$-optimal policy and the theoretical lower bounds. For fixed values of $\epsilon$, which mostly occur in real applications, our algorithm improves upon the previous best sample and time complexity bounds by a factor of $(1 - \gamma)^{-1}$ where $\gamma \in (0, 1)$, the discount factor, is typically close to 1.

We achieve our results by combining and strengthening techniques from both [AMK13] and [SWWY18]. On the one hand, in [AMK13] the authors showed that simply constructing a "sparsified" MDP model by taking samples and then solving this model to high precision yields a sample optimal algorithm in our setting for computing the approximate value of every state. On the other hand, [SWWY18] provided faster algorithms for solving explicit DMDPs and improved sample and time complexities given a sampling oracle. In fact, as we show in Appendix B.1, simply combining these two results yields the first nearly optimal runtime for approximately learning the value function with a generative model. Unfortunately, it is known that an approximate-optimal value function does not immediately yield an approximate-optimal policy of comparable quality (see e.g. [Ber13]) and it is was previously unclear how to combine these methods to improve upon previous known bounds for computing an approximate policy. To achieve our policy computation algorithm we therefore

open up both the algorithms and the analysis in [AMK13] and [SWWY18], combining them in non-trivial ways. Our proofs leverage techniques ranging from standard probabilistic analysis tools such as Hoeffding and Bernstein inequalities, to optimization techniques such as variance reduction, to properties specific to MDPs such as the Bellman fixed-point recursion for expectation and variance of the optimal value vector, and monotonicity of value iteration.

Finally, we extend our method to finite-horizon MDPs, which are also occurred frequently in real applications. We show that the number of samples needed by this algorithm is $\widetilde{O}(H^3|\mathcal{S}||\mathcal{A}|\epsilon^{-2})$, in order to obtain an $\epsilon$-optimal policy for $H$-horizon MDP (see Appendix F). We also show that the preceding sample complexity is optimal up to logarithmic factors by providing a matching lower bound. We hope this work ultimately opens the door for future practical and theoretical work on solving MDPs and efficient RL more broadly.

## 2 Preliminaries

We use calligraphy upper case letters for sets or operators, e.g., $\mathcal{S}$, $\mathcal{A}$ and $\mathcal{T}$. We use bold small case letters for vectors, e.g., $\boldsymbol{v}, \boldsymbol{r}$. We denote $\boldsymbol{v}_s$ or $\boldsymbol{v}(s)$ as the $s$-th entry of vector $\boldsymbol{v}$. We denote matrix as bold upper case letters, e.g., $\boldsymbol{P}$. We denote constants as normal upper case letters, e.g., $M$. For a vector $\boldsymbol{v} \in \mathbb{R}^{\mathcal{N}}$ for index set $\mathcal{N}$, we denote $\sqrt{\boldsymbol{v}}$, $|\boldsymbol{v}|$, and $\boldsymbol{v}^2$ vectors in $\mathbb{R}^{\mathcal{N}}$ with $\sqrt{\cdot}$, $|\cdot|$, and $(\cdot)^2$ acting coordinate-wise. For two vectors $\boldsymbol{v}, \boldsymbol{u} \in \mathbb{R}^{\mathcal{N}}$, we denote by $\boldsymbol{v} \leq \boldsymbol{u}$ as coordinate-wise comparison, i.e., $\forall i \in \mathcal{N} : \boldsymbol{v}(i) \leq \boldsymbol{u}(i)$. The same definition are defined to relations $\leq, <$ and $>$.

We describe a DMDP by the tuple $(\mathcal{S}, \mathcal{A}, \boldsymbol{P}, \boldsymbol{r}, \gamma)$, where $\mathcal{S}$ is a finite state space, $\mathcal{A}$ is a finite action space, $\boldsymbol{P} \in \mathbb{R}^{\mathcal{S} \times \mathcal{A} \times \mathcal{S}}$ is the state-action-state transition matrix, $\boldsymbol{r} \in \mathbb{R}^{\mathcal{S} \times \mathcal{A}}$ is the state-action reward vector, and $\gamma \in (0, 1)$ is a discount factor. We use $\boldsymbol{P}_{s,a}(s')$ to denote the probability of going to state $s'$ from state $s$ when taking action $a$. We also identify each $\boldsymbol{P}_{s,a}$ as a vector in $\mathbb{R}^S$. We use $\boldsymbol{r}_{s,a}$ to denote the reward obtained from taking action $a \in \mathcal{A}$ at state $s \in \mathcal{S}$ and assume $\boldsymbol{r} \in [0, 1]^{\mathcal{S} \times \mathcal{A}}$.[3] For a vector $\boldsymbol{v} \in \mathbb{R}^S$, we denote $\boldsymbol{P}\boldsymbol{v} \in \mathbb{R}^{\mathcal{S} \times \mathcal{A}}$ as $(\boldsymbol{P}\boldsymbol{v})_{s,a} = \boldsymbol{P}_{s,a}^{\top}\boldsymbol{v}$. A policy $\pi : \mathcal{S} \to \mathcal{A}$ maps each state to an action. The objective of MDP is to find the optimal policy $\pi^*$ that maximizes the expectation of the cumulative sum of discounted rewards.

In the remainder of this section we give definitions for several prominent concepts in MDP analysis that we use throughout the paper.

**Definition 2.1** (Bellman Value Operator). For a given DMDP the *value operator* $\mathcal{T} : \mathbb{R}^S \mapsto \mathbb{R}^S$ is defined for all $\boldsymbol{u} \in \mathbb{R}^S$ and $s \in \mathcal{S}$ by $\mathcal{T}(\boldsymbol{u})_s = \max_{a \in \mathcal{A}}[\boldsymbol{r}_a(s) + \gamma \cdot \boldsymbol{P}_{s,a}^{\top}\boldsymbol{v}]$, and we let $\boldsymbol{v}^*$ denote the *value of the optimal policy* $\pi^*$, which is the unique vector such that $\mathcal{T}(\boldsymbol{v}^*) = \boldsymbol{v}^*$.

**Definition 2.2** (Policy). We call any vector $\pi \in \mathcal{A}^S$ a *policy* and say that the action prescribed by policy $\pi$ to be taken at state $s \in \mathcal{S}$ is $\pi_s$. We let $\mathcal{T}_{\pi} : \mathbb{R}^S \mapsto \mathbb{R}^S$ denote the *value operator associated with* $\pi$ defined for all $\boldsymbol{u} \in \mathbb{R}^S$ and $s \in \mathcal{S}$ by $\mathcal{T}_{\pi}(\boldsymbol{u})_s = \boldsymbol{r}_{s,\pi(s)} + \gamma \cdot \boldsymbol{P}_{s,\pi(s)}^{\top}\boldsymbol{u}$ , and we let $\boldsymbol{v}^{\pi}$ denote the *values of policy* $\pi$, which is the unique vector such that $\mathcal{T}_{\pi}(\boldsymbol{v}^{\pi}) = \boldsymbol{v}^{\pi}$.

Note that $\mathcal{T}_{\pi}$ can be viewed as the value operator for the modified MDP where the only available action from each state is given by the policy $\pi$. Note that this modified MDP is essentially just an uncontrolled Markov Chain, i.e. there are no action choices that can be made.

**Definition 2.3** ($\epsilon$-optimal value and policy). We say values $\boldsymbol{u} \in \mathbb{R}^S$ are $\epsilon$-*optimal* if $\|\boldsymbol{v}^* - \boldsymbol{u}\|_{\infty} \leq \epsilon$ and policy $\pi \in \mathcal{A}^S$ is $\epsilon$-*optimal* if $\|\boldsymbol{v}^* - \boldsymbol{v}^{\pi}\|_{\infty} \leq \epsilon$, i.e. the values of $\pi$ are $\epsilon$-optimal.

**Definition 2.4** (Q-function). For any policy $\pi$, we define the Q-function of a MDP with respect to $\pi$ as a vector $\boldsymbol{Q} \in \mathbb{R}^{\mathcal{S} \times \mathcal{A}}$ such that $\boldsymbol{Q}^{\pi}(s, a) = \boldsymbol{r}(s, a) + \gamma \boldsymbol{P}_{s,a}^{\top}\boldsymbol{v}^{\pi}$. The optimal Q-function is defined as $\boldsymbol{Q}^* = \boldsymbol{Q}^{\pi^*}$. We call any vector $\boldsymbol{Q} \in \mathbb{R}^{\mathcal{S} \times \mathcal{A}}$ a Q-function even though it may not relate to a policy or a value vector and define $\boldsymbol{v}(\boldsymbol{Q}) \in \mathbb{R}^S$ and $\pi(\boldsymbol{Q}) \in \mathcal{A}^S$ as the value and policy implied by $\boldsymbol{Q}$, by

$$\forall s \in \mathcal{S} : \boldsymbol{v}(\boldsymbol{Q})(s) = \max_{a \in \mathcal{A}} \boldsymbol{Q}(s, a) \quad \text{and} \quad \pi(\boldsymbol{Q})(s) = \arg\max_{a \in \mathcal{A}} \boldsymbol{Q}(s, a).$$

For a policy $\pi$, let $\boldsymbol{P}^{\pi}\boldsymbol{Q} \in \mathbb{R}^{\mathcal{S} \times \mathcal{A}}$ be defined as $(\boldsymbol{P}^{\pi}\boldsymbol{Q})(s, a) = \sum_{s' \in \mathcal{S}} \boldsymbol{P}_{s,a}(s')\boldsymbol{Q}(s', \pi(s'))$.

# 3   Technique Overview

In this section we provide a more detailed and technical overview of our approach. At a high level, our algorithm shares a similar framework as the variance reduction algorithm presented in [SWWY18]. This algorithm used two crucial algorithmic techniques, which are also critical in this paper. We call these techniques as the *monotonicity* technique and the *variance reduction* technique. Our algorithm and the results of this paper can be viewed as an advanced, non-trivial integration of these two methods, augmented with a third technique which we refer to as a *total-variation* technique which was discovered in several papers [MM99, LH12, AMK13]. In the remainder of this section we give an overview of these techniques and through this, explain our algorithm.

**The Monotonicity Technique**   Recall that the classic value iteration algorithm for solving a MDP repeatedly applies the following rule

$$\boldsymbol{v}^{(i)}(s) \leftarrow \max_a (r(s,a) + \gamma \boldsymbol{P}_{s,a}^\top \boldsymbol{v}^{(i-1)}). \tag{3.1}$$

A greedy policy $\pi^{(i)}$ can be obtained at each iteration $i$ by

$$\forall s : \pi^{(i)}(s) \leftarrow \operatorname{argmax}_a (r(s,a) + \gamma \boldsymbol{P}_{s,a}^\top \boldsymbol{v}^{(i)}). \tag{3.2}$$

For any $u > 0$, it can be shown that if one can approximate $\boldsymbol{v}^{(i)}(s)$ with $\widehat{\boldsymbol{v}}^{(i)}(s)$ such that $\|\widehat{\boldsymbol{v}}^{(i)} - \boldsymbol{v}^{(i)}\|_\infty \leq (1-\gamma)u$ and run the above value iteration algorithm using these approximated values, then after $\Theta((1-\gamma)^{-1} \log[u^{-1}(1-\gamma)^{-1}])$ iterations, the final iteration gives an value function that is $u$-optimal ([Ber13]). However, a $u$-optimal value function only yields a $u/(1-\gamma)$-optimal greedy policy (in the worst case), even if (3.2) is precisely computed. To get around this additional loss, a monotone-VI algorithm was proposed in [SWWY18] as follows. At each iteration, this algorithm maintains not only an approximated value $\boldsymbol{v}^{(i)}$ but also a policy $\pi^{(i)}$. The key for improvement is to keep values as a lower bound of the value of the policy on a set of sample paths with high probability. In particular, the following *monotonicity condition* was maintained with high probability

$$\boldsymbol{v}^{(i)} \leq \mathcal{T}_{\pi^{(i)}}(\boldsymbol{v}^{(i)}) .$$

By the monotonicity of the Bellman's operator, the above equation guarantees that $\boldsymbol{v}^{(i)} \leq \boldsymbol{v}^{\pi^{(i)}}$. If this condition is satisfied, then, if after $R$ iterations of approximate value iteration we obtain an value $\widehat{\boldsymbol{v}}^{(R)}$ that is $u$-optimal then we also obtain a policy $\pi^{(R)}$ which by the monotonicity condition and the monotonicity of the Bellman operator $\mathcal{T}_{\pi^{(R)}}$ yields

$$\boldsymbol{v}^{(R)} \leq \mathcal{T}_{\pi^{(R)}}(\boldsymbol{v}^{(R)}) \leq \mathcal{T}_{\pi^{(R)}}^2(\boldsymbol{v}^{(R)}) \leq \ldots \leq \mathcal{T}_{\pi^{(R)}}^\infty(\boldsymbol{v}^{(R)}) = \boldsymbol{v}^{\pi^{(R)}} \leq \boldsymbol{v}^*.$$

and therefore this $\pi^{(R)}$ is an $u$-optimal policy. Ultimately, this technique avoids the standard loss of a $(1-\gamma)^{-1}$ factor when converting values to policies.

**The Variance Reduction Technique**   Suppose now that we provide an algorithm that maintains the monotonicity condition using random samples from $\boldsymbol{P}_{s,a}$ to approximately compute (3.1). Further, suppose we want to obtain a new value function and policy that is at least $(u/2)$-optimal. In order to obtain the desired accuracy, we need to approximate $\boldsymbol{P}_{s,a}^\top \boldsymbol{v}^{(i)}$ up to error at most $(1-\gamma)u/2$. Since $\|\boldsymbol{v}^{(i)}\|_\infty \leq (1-\gamma)^{-1}$, by Hoeffding bound, $\widetilde{O}((1-\gamma)^{-4}u^{-2})$ samples suffices. Note that the number of samples also determines the computation time and therefore each iteration takes $\widetilde{O}((1-\gamma)^{-4}u^{-2}|\mathcal{S}||\mathcal{A}|)$ samples/computation time and $\widetilde{O}((1-\gamma)^{-1})$ iterations for the value iteration to converge. Overall, this yields a sample/computation complexity of $\widetilde{O}((1-\gamma)^{-5}u^{-2}|\mathcal{S}||\mathcal{A}|)$. To reduce the $(1-\gamma)^{-5}$ dependence, [SWWY18] uses properties of the input (and the initialization) vectors: $\|\boldsymbol{v}^{(0)} - \boldsymbol{v}^*\|_\infty \leq u$ and rewrites value iteration (3.1) as follows

$$\boldsymbol{v}^{(i)}(s) \leftarrow \max_a \big[ r(s,a) + \boldsymbol{P}_{s,a}^\top (\boldsymbol{v}^{(i-1)} - \boldsymbol{v}^{(0)}) + \boldsymbol{P}_{s,a}^\top \boldsymbol{v}^{(0)} \big], \tag{3.3}$$

Notice that $\boldsymbol{P}_{s,a}^\top \boldsymbol{v}^{(0)}$ is shared over all iterations and we can approximate it up to error $(1-\gamma)u/4$ using only $\widetilde{O}((1-\gamma)^{-4}u^{-2})$ samples. For every iteration, we have $\|\boldsymbol{v}^{(i-1)} - \boldsymbol{v}^{(0)}\|_\infty \leq u$ (recall that we demand the monotonicity is satisfied at each iteration). Hence $\boldsymbol{P}_{s,a}^\top (\boldsymbol{v}^{(i-1)} - \boldsymbol{v}^{(0)})$ can be approximated up to error $(1-\gamma)u/4$ using only $\widetilde{O}((1-\gamma)^{-2})$ samples (note that there is no $u$-dependence here). By this technique, over $\widetilde{O}((1-\gamma)^{-1})$ iterations only $\widetilde{O}((1-\gamma)^{-4}u^{-2} + (1-\gamma)^{-3})$ samples/computation per state action pair are needed, i.e. there is a $(1-\gamma)$ improvement.

**The Total-Variance Technique**    By combining the monotonicity technique and variance reduction technique, one can obtain a $\widetilde{O}((1-\gamma)^{-4})$ sample/running time complexity (per state-action pair) on computing a policy; this was one of the results [SWWY18]. However, there is a gap between this bound and the best known lower bound of $\widetilde{\Omega}[|S||A|\epsilon^{-2}(1-\gamma)^{-3}]$ [AMK13]. Here we show how to remove the last $(1-\gamma)$ factor by better exploiting the structure of the MDP. In [SWWY18] the update error in each iteration was set to be at most $(1-\gamma)u/2$ to compensate for error accumulation through a horizon of length $(1-\gamma)^{-1}$ (i.e., the accumulated error is sum of the estimation error at each iteration). To improve we show how to leverage previous work to show that the true error accumulation is much less. To see this, let us now switch to Bernstein inequality. Suppose we would like to estimate the value function of some policy $\pi$. The estimation error vector of the value function is upper bounded by $\widetilde{O}(\sqrt{\boldsymbol{\sigma}_\pi/m})$, where $\boldsymbol{\sigma}_\pi(s) = \mathrm{Var}_{s' \sim \boldsymbol{P}_{s,\pi(s)}}(\boldsymbol{v}^\pi(s'))$ denotes the variance of the value of the next state if starting from state $s$ by playing policy $\pi$, and $m$ is the number of samples collected per state-action pair. The accumulated error due to estimating value functions can be shown to obey the following inequality (upper to logarithmic factors)

$$\text{accumated error} \propto \sum_{i=0}^\infty \gamma^i \boldsymbol{P}_\pi^i \sqrt{\boldsymbol{\sigma}_\pi/m} \leq c_1 \left( \frac{1}{1-\gamma} \sum_{i=0}^\infty \gamma^{2i} \boldsymbol{P}_\pi^i \boldsymbol{\sigma}_\pi/m \right)^{1/2},$$

where $c_1$ is a constant and the inequality follows from a Cauchy-Swartz-like inequality. According to the *law of total variance*, for any given policy $\pi$ (in particular, the optimal policy $\pi^*$) and initial state $s$, the expected sum of variance of the tail sums of rewards, $\sum \gamma^{2i} \boldsymbol{P}_\pi^i \boldsymbol{\sigma}_\pi$, is exactly the variance of the total return by playing the policy $\pi$. This observation was previously used in the analysis of [MM99, LH12, AMK13]. Since the upper bound on the total return is $(1-\gamma)^{-1}$, it can be shown that $\sum_i \gamma^{2i} \boldsymbol{P}_\pi^i \boldsymbol{\sigma}_\pi \leq (1-\gamma)^{-2} \cdot \boldsymbol{1}$ and therefore the total error accumulation is $\sqrt{(1-\gamma)^{-3}/m}$. Thus picking $m \approx (1-\gamma)^{-3}\epsilon^{-2}$ is sufficient to control the accumulated error (instead of $(1-\gamma)^{-4}$). To analyze our algorithm, we will apply the above inequality to the optimal policy $\pi^*$ to obtain our final error bound.

**Putting it All Together**    In the next section we show how to combine these three techniques into one algorithm and make them work seamlessly. In particular, we provide and analyze how to combine these techniques into an Algorithm 1 which can be used to at least halve the error of a current policy. Applying this routine a logarithmic number of time then yields our desired bounds. In the input of the algorithm, we demand the input value $\boldsymbol{v}^{(0)}$ and $\pi^{(0)}$ satisfies the required monotonicity requirement, i.e., $\boldsymbol{v}^{(0)} \leq \mathcal{T}_{\pi^{(0)}}(\boldsymbol{v}^{(0)})$ (in the first iteration, the zero vector $\boldsymbol{0}$ and an arbitrary policy $\pi$ satisfies the requirement). We then pick a set of samples to estimate $\boldsymbol{P}\boldsymbol{v}^{(0)}$ accurately with $\widetilde{O}((1-\gamma)^{-3}\epsilon^{-2})$ samples per state-action pair. The same set of samples is used to estimate the variance vector $\boldsymbol{\sigma}_{\boldsymbol{v}^*}$. These estimates serve as the initialization of the algorithm. In each iteration $i$, we draw fresh new samples to compute estimate of $\boldsymbol{P}(\boldsymbol{v}^{(i)} - \boldsymbol{v}^{(0)})$. The sum of the estimate of $\boldsymbol{P}\boldsymbol{v}^{(0)}$ and $\boldsymbol{P}(\boldsymbol{v}^{(i)} - \boldsymbol{v}^{(0)})$ gives an estimate of $\boldsymbol{P}\boldsymbol{v}^{(i)}$. We then make the above estimates have one-sided error by shifting them according to their estimation errors (which is estimated from the Bernstein inequality). These one-side error estimates allow us to preserve monotonicity, i.e., guarantees the new value is always improving on the entire sample path with high probability. The estimate of $\boldsymbol{P}\boldsymbol{v}^{(i)}$ is plugged in to the Bellman's operator and gives us new value function, $\boldsymbol{v}^{(i+1)}$ and policy $\pi^{(i+1)}$, satisfying the monotonicity and advancing accuracy. Repeating the above procedure for the desired number of iterations completes the algorithm.

## 4    Algorithm and Analysis

In this section we provide and analyze our near sample/time optimal $\epsilon$-policy computation algorithm. As discussed in Section 3 our algorithm combines three main ideas: variance reduction, the monotone value/policy iteration, and the reduction of accumulated error via Bernstein inequality. These ingredients are used in the Algorithm 1 to provide a routine which halves the error of a given policy. We analyze this procedure in Section 4.1 and use it to obtain our main result in Section 4.2.

### 4.1    The Analysis of the Variance Reduced Algorithm

In this section we analyze Algorithm 1, showing that each iteration of the algorithm approximately contracts towards the optimal value and policy and that ultimately the algorithm halves the error

---

**Algorithm 1** Variance-Reduced QVI

---

1: **Input:** A sampling oracle for DMDP $\mathcal{M} = (\mathcal{S}, \mathcal{A}, \boldsymbol{r}, \boldsymbol{P}, \gamma)$
2: **Input:** Upper bound on error $u \in [0, (1-\gamma)^{-1}]$ and error probability $\delta \in (0,1)$
3: **Input:** Initial values $\boldsymbol{v}^{(0)}$ and policy $\pi^{(0)}$ such that $\boldsymbol{v}^{(0)} \le \mathcal{T}_{\pi^{(0)}}\boldsymbol{v}^{(0)}$, and $\boldsymbol{v}^* - \boldsymbol{v}^{(0)} \le u\mathbf{1}$;
4: **Output:** $\boldsymbol{v}, \pi$ such that $\boldsymbol{v} \le \mathcal{T}_\pi(\boldsymbol{v})$ and $\boldsymbol{v}^* - \boldsymbol{v} \le (u/2) \cdot \mathbf{1}$.
5:
6: **INITIALIZATION:**
7: Let $\beta \leftarrow (1-\gamma)^{-1}$, and $R \leftarrow \lceil c_1\beta \ln[\beta u^{-1}]\rceil$ for constant $c_1$;
8: Let $m_1 \leftarrow c_2\beta^3 u^{-2}\log(8|\mathcal{S}||\mathcal{A}|\delta^{-1})$ for constant $c_2$;
9: Let $m_2 \leftarrow c_3\beta^2 \log[2R|\mathcal{S}||\mathcal{A}|\delta^{-1}]$ for constant $c_3$;
10: Let $\alpha_1 \leftarrow m_1^{-1}\log(8|\mathcal{S}||\mathcal{A}|\delta^{-1})$;
11: For each $(s,a) \in \mathcal{S} \times \mathcal{A}$, sample independent samples $s_{s,a}^{(1)}, s_{s,a}^{(2)}, \ldots, s_{s,a}^{(m_1)}$ from $\boldsymbol{P}_{s,a}$;
12: Initialize $\boldsymbol{w} = \widetilde{\boldsymbol{w}} = \widehat{\boldsymbol{\sigma}} = \boldsymbol{Q}^{(0)} \leftarrow \mathbf{0}_{\mathcal{S}\times\mathcal{A}}$, and $i \leftarrow 0$;
13: **for** each $(s,a) \in \mathcal{S} \times \mathcal{A}$ **do**
14:     \\*Compute empirical estimates of* $\boldsymbol{P}_{s,a}^\top \boldsymbol{v}^{(0)}$ *and* $\boldsymbol{\sigma}_{\boldsymbol{v}^{(0)}}(s,a)$
15:     Let $\widetilde{\boldsymbol{w}}(s,a) \leftarrow \frac{1}{m_1}\sum_{j=1}^{m_1} \boldsymbol{v}^{(0)}(s_{s,a}^{(j)})$
16:     Let $\widehat{\boldsymbol{\sigma}}(s,a) \leftarrow \frac{1}{m_1}\sum_{j=1}^{m_1}(\boldsymbol{v}^{(0)})^2(s_{s,a}^{(j)}) - \widetilde{\boldsymbol{w}}^2(s,a)$
17:
18:     \\*Shift the empirical estimate to have one-sided error and guarantee monotonicity*
19:     $\boldsymbol{w}(s,a) \leftarrow \widetilde{\boldsymbol{w}}(s,a) - \sqrt{2\alpha_1\widehat{\boldsymbol{\sigma}}(s,a)} - 4\alpha_1^{3/4}\|\boldsymbol{v}^{(0)}\|_\infty - (2/3)\alpha_1\|\boldsymbol{v}^{(0)}\|_\infty$
20:
21:     \\*Compute coarse estimate of the Q-function*
22:     $\boldsymbol{Q}^{(0)}(s,a) \leftarrow \boldsymbol{r}(s,a) + \gamma\boldsymbol{w}(s,a)$
23:
24: **REPEAT:**       \\*successively improve*
25: **for** $i = 1$ to $R$ **do**
26:     \\*Compute* $\boldsymbol{g}^{(i)}$ *the estimate of* $\boldsymbol{P}[\boldsymbol{v}^{(i)} - \boldsymbol{v}^{(0)}]$ *with one-sided error*
27:     Let $\boldsymbol{v}^{(i)} \leftarrow \boldsymbol{v}(\boldsymbol{Q}^{(i-1)})$, $\pi^{(i)} \leftarrow \pi(\boldsymbol{Q}^{(i-1)})$; \\*let* $\widetilde{\boldsymbol{v}}^{(i)} \leftarrow \boldsymbol{v}^{(i)}, \widetilde{\pi}^{(i)} \leftarrow \pi^{(i)}$ *(for analysis)*;
28:     For each $s \in \mathcal{S}$, if $\boldsymbol{v}^{(i)}(s) \le \boldsymbol{v}^{(i-1)}(s)$, then $\boldsymbol{v}^{(i)}(s) \leftarrow \boldsymbol{v}^{(i-1)}(s)$ and $\pi^{(i)}(s) \leftarrow \pi^{(i-1)}(s)$;
29:     For each $(s,a) \in \mathcal{S} \times \mathcal{A}$, draw independent samples $\widetilde{s}_{s,a}^{(1)}, \widetilde{s}_{s,a}^{(2)}, \ldots, \widetilde{s}_{s,a}^{(m_2)}$ from $\boldsymbol{P}_{s,a}$;
30:     Let $\boldsymbol{g}^{(i)}(s,a) \leftarrow \frac{1}{m_2}\sum_{j=1}^{m_2}\left[\boldsymbol{v}^{(i)}(\widetilde{s}_{s,a}^{(j)}) - \boldsymbol{v}^{(0)}(\widetilde{s}_{s,a}^{(j)})\right] - (1-\gamma)u/8$;
31:
32:     \\*Improve* $\boldsymbol{Q}^{(i)}$
33:     $\boldsymbol{Q}^{(i)} \leftarrow \boldsymbol{r} + \gamma \cdot [\boldsymbol{w} + \boldsymbol{g}^{(i)}]$;
34: **return** $\boldsymbol{v}^{(R)}, \pi^{(R)}$.

---

of the input value and policy with high probability. All proofs in this section are deferred to Appendix E.1.

We start with bounding the error of $\widetilde{\boldsymbol{w}}$ and $\widehat{\boldsymbol{\sigma}}$ defined in Line 15 and 16 of Algorithm 1. Notice that these are the empirical estimations of $\boldsymbol{P}_{s,a}^\top\boldsymbol{v}^{(0)}$ and $\boldsymbol{\sigma}_{\boldsymbol{v}^{(0)}}(s,a)$.

**Lemma 4.1** (Empirical Estimation Error). *Let $\widetilde{\boldsymbol{w}}$ and $\widehat{\boldsymbol{\sigma}}$ be computed in Line 15 and 16 of Algorithm 1. Recall that $\widetilde{\boldsymbol{w}}$ and $\widehat{\boldsymbol{\sigma}}$ are empirical estimates of $\boldsymbol{P}\boldsymbol{v}$ and $\boldsymbol{\sigma}_{\boldsymbol{v}} = \boldsymbol{P}\boldsymbol{v}^2 - (\boldsymbol{P}\boldsymbol{v})^2$ using $m_1$ samples per $(s,a)$ pair. With probability at least $1 - \delta$, for $L \stackrel{\text{def}}{=} \log(8|\mathcal{S}||\mathcal{A}|\delta^{-1})$, we have*

$$\left|\widetilde{\boldsymbol{w}} - \boldsymbol{P}^\top\boldsymbol{v}^{(0)}\right| \le \sqrt{2m_1^{-1}\boldsymbol{\sigma}_{\boldsymbol{v}^{(0)}} \cdot L} + 2(3m_1)^{-1}\|\boldsymbol{v}^{(0)}\|_\infty L \tag{4.1}$$

*and*

$$\forall (s,a) \in \mathcal{S} \times \mathcal{A}: \quad \left|\widehat{\boldsymbol{\sigma}}(s,a) - \boldsymbol{\sigma}_{\boldsymbol{v}^{(0)}}(s,a)\right| \le 4\|\boldsymbol{v}^{(0)}\|_\infty^2 \cdot \sqrt{2m_1^{-1}L}. \tag{4.2}$$

The proof is a straightforward application of Bernstein's inequality and Hoeffding's inequality.

Next we show that the difference between $\sigma_{\boldsymbol{v}^{(0)}}$ and $\sigma_{\boldsymbol{v}^*}$ is also bounded.

**Lemma 4.2.** *Suppose $\|\boldsymbol{v} - \boldsymbol{v}^*\|_\infty \leq \epsilon$ for some $\epsilon > 0$, then $\sqrt{\boldsymbol{\sigma_v}} \leq \sqrt{\boldsymbol{\sigma_{v^*}}} + \epsilon \cdot \mathbf{1}$.*

Next we show that in Line 30, the computed $\boldsymbol{g}^{(i)}$ concentrates to and is an overestimate of $\boldsymbol{P}[\boldsymbol{v}^{(i)} - \boldsymbol{v}^{(0)}]$ with high probability.

**Lemma 4.3.** *Let $\boldsymbol{g}^{(i)}$ be the estimate of $\boldsymbol{P}\big[\boldsymbol{v}^{(i)} - \boldsymbol{v}^{(0)}\big]$ defined in Line 30 of Algorithm 1. Then conditioning on the event that $\|\boldsymbol{v}^{(i)} - \boldsymbol{v}^{(0)}\|_\infty \leq 2u$, with probability at least $1 - \delta/R$,*

$$\boldsymbol{P}\big[\boldsymbol{v}^{(i)} - \boldsymbol{v}^{(0)}\big] - \frac{(1-\gamma)u}{4} \cdot \mathbf{1} \leq \boldsymbol{g}^{(i)} \leq \boldsymbol{P}\big[\boldsymbol{v}^{(i)} - \boldsymbol{v}^{(0)}\big]$$

*provided appropriately chosen constants $c_1$, $c_2$, and $c_3$ in Algorithm 1.*

Now we present the key contraction lemma, in which we set the constants, $c_1, c_2, c_3$, in Algorithm 1 to be sufficiently large (e.g., $c_1 \geq 4, c_2 \geq 8192, c_3 \geq 128$). Note that these constants only need to be sufficiently large so that the concentration inequalities hold.

**Lemma 4.4.** *Let $\boldsymbol{Q}^{(i)}$ be the estimated Q-function of $\boldsymbol{v}^{(i)}$ in Line 33 of Algorithm 1. Let $\pi^{(i)}$ and $\boldsymbol{v}^{(i)}$ be estimated in iteration $i$, as defined in Line 27 and 28. Then, with probability at least $1 - 2\delta$, for all $1 \leq i \leq R$,*

$$\boldsymbol{v}^{(i-1)} \leq \boldsymbol{v}^{(i)} \leq \mathcal{T}_{\pi^{(i)}}[\boldsymbol{v}^{(i)}], \quad \boldsymbol{Q}^{(i)} \leq \boldsymbol{r} + \gamma \boldsymbol{P} \boldsymbol{v}^{(i)}, \quad and \quad \boldsymbol{Q}^* - \boldsymbol{Q}^{(i)} \leq \gamma \boldsymbol{P}^{\pi^*}\big[\boldsymbol{Q}^* - \boldsymbol{Q}^{(i-1)}\big] + \boldsymbol{\xi},$$

*where for $\alpha_1 = m_1^{-1}L < 1$ the error vector $\boldsymbol{\xi}$ satisfies*

$$\mathbf{0} \leq \boldsymbol{\xi} \leq C\sqrt{\alpha_1 \boldsymbol{\sigma_{v^*}}} + \Big[(1-\gamma)u/C + C\alpha_1^{3/4}\|\boldsymbol{v}^{(0)}\|_\infty\Big] \cdot \mathbf{1},$$

*for some sufficiently large constant $C \geq 8$.*

Using the previous lemmas we can prove the guarantees of Algorithm 1.

**Proposition 4.5.** *On an input value vector $\boldsymbol{v}^{(0)}$, policy $\pi^{(0)}$, and parameters $u \in (0, (1-\gamma)^{-1}], \delta \in (0,1)$ such that $\boldsymbol{v}^{(0)} \leq \mathcal{T}_{\pi^{(0)}}[\boldsymbol{v}^{(0)}]$, and $\boldsymbol{v}^* - \boldsymbol{v}^{(0)} \leq u\mathbf{1}$, Algorithm 1 halts in time $O((1-\gamma)^{-1}u^{-2} \cdot |\mathcal{S}||\mathcal{A}| \cdot \log(|\mathcal{S}||\mathcal{A}|\delta^{-1}(1-\gamma)^{-1}u^{-1}))$ and outputs values $\boldsymbol{v}$ and policy $\pi$ such that $\boldsymbol{v} \leq \mathcal{T}_\pi(\boldsymbol{v})$ and $\boldsymbol{v}^* - \boldsymbol{v} \leq (u/2)\mathbf{1}$ with probability at least $1 - \delta$, provided appropriately chosen constants, $c_1, c_2, c_3$.*

We prove this proposition by iteratively applying Lemma 4.4. Suppose $\boldsymbol{v}^{(R)}$ is the output of the algorithm, after $R$ iterations. We show $\boldsymbol{v}^* - \boldsymbol{v}^{(R)} \leq \gamma^{R-1}\boldsymbol{P}^{\pi^*}\big[\boldsymbol{Q}^* - \boldsymbol{Q}^0\big] + (\boldsymbol{I} - \gamma\boldsymbol{P}^{\pi^*})^{-1}\boldsymbol{\xi}$. Notice that $(\boldsymbol{I} - \gamma\boldsymbol{P}^{\pi^*})^{-1}\boldsymbol{\xi}$ is related to $(\boldsymbol{I} - \gamma\boldsymbol{P}^{\pi^*})^{-1}\sqrt{\boldsymbol{\sigma_{v^*}}}$. We then apply the variance analytical tools presented in Section C to show that $(\boldsymbol{I} - \gamma\boldsymbol{P}^{\pi^*})^{-1}\boldsymbol{\xi} \leq (u/4)\mathbf{1}$ when setting the constants properly in Algorithm 1. We refer this technique as the *total-variance technique*, since $\|(\boldsymbol{I} - \gamma\boldsymbol{P}^{\pi^*})^{-1}\sqrt{\boldsymbol{\sigma_{v^*}}}\|_\infty^2 \leq O[(1-\gamma)^{-3}]$ instead of a naïve bound of $(1-\gamma)^{-4}$. We complete the proof by choosing $R = \widetilde{\Theta}((1-\gamma)^{-1}\log(u^{-1}))$ and showing that $\gamma^{R-1}\boldsymbol{P}^{\pi^*}\big[\boldsymbol{Q}^* - \boldsymbol{Q}^0\big] \leq (u/4)\mathbf{1}$.

### 4.2 From Halving the Error to Arbitrary Precision

In the previous section, we provided an algorithm that on an input policy, outputs a policy with value vector that has $\ell_\infty$ distance to the optimal value vector only half of that of the input one. In this section, we give a complete policy computation algorithm by by showing that it is possible to apply this error "halving" procedure iteratively. We summarize our meta algorithm in Algorithm 2. Note that in the algorithm, each call of HALFERR draws new samples from the sampling oracle. We refer in this section to Algorithm 1 as a subroutine HALFERR, which given an input MDP $\mathcal{M}$ with a sampling oracle, an input value function $\boldsymbol{v}^{(i)}$, and an input policy $\pi^{(i)}$, outputs an value function $\boldsymbol{v}^{(i+1)}$ and a policy $\pi^{(i+1)}$.

Combining Algorithm 2 and Algorithm 1, we are ready to present main result.

**Theorem 4.6.** *Let $\mathcal{M} = (\mathcal{S}, \mathcal{A}, \boldsymbol{P}, \boldsymbol{r}, \gamma)$ be a DMDP with a generative model. Suppose we can sample a state from each probability vector $\boldsymbol{P}_{s,a}$ within time $O(1)$. Then for any $\epsilon, \delta \in (0,1)$, there exists an algorithm that halts in time*

$$T := O\left[\frac{|\mathcal{S}||\mathcal{A}|}{(1-\gamma)^3\epsilon^2}\log\left(\frac{|\mathcal{S}||\mathcal{A}|}{(1-\gamma)\delta\epsilon}\right)\log\left(\frac{1}{(1-\gamma)\epsilon}\right)\right]$$

---

**Algorithm 2** Meta Algorithm

---
1: **Input:** A sampling oracle of some $\mathcal{M} = (\mathcal{S}, \mathcal{A}, \boldsymbol{r}, \boldsymbol{P}, \gamma), \epsilon > 0, \delta \in (0, 1)$
2: **Initialize:** $\boldsymbol{v}^{(0)} \leftarrow \boldsymbol{0}, \pi^{(0)} \leftarrow$ arbitrary policy, $R \leftarrow \Theta[\log(\epsilon^{-1}(1-\gamma)^{-1})]$
3: **for** $i = \{1, 2, \ldots, R\}$ **do**
4:     //HALFERR *is initialized with* $QVI(u = 2^{-i+1}(1-\gamma)^{-1}, \delta, \boldsymbol{v}^{(0)} = \boldsymbol{v}^{(i-1)}, \pi^{(0)} = \pi^{(i-1)})$
5:     $\boldsymbol{v}^{(i)}, \pi^{(i)} \leftarrow$ HALFERR $\leftarrow \boldsymbol{v}^{(i-1)}, \pi^{(i-1)}$
6: **Output:** $\boldsymbol{v}^{(R)}, \pi^{(R)}$.

---

*and obtains a policy $\pi$ such that $\boldsymbol{v}^* - \epsilon\boldsymbol{1} \leq \boldsymbol{v}^\pi \leq \boldsymbol{v}^*$, with probability at least $1 - \delta$ where $\boldsymbol{v}^*$ is the optimal value of $\mathcal{M}$. The algorithm uses space $O(|\mathcal{S}||\mathcal{A}|)$ and queries the generative model for at most $O(T)$ fresh samples.*

*Remark* 4.7. The full analysis of the halving algorithm is presented in Section E.2. Our algorithm can be implemented in space $O(|\mathcal{S}||\mathcal{A}|)$ since in Algorithm 1, the initialization phase can be done for each $(s, a)$ and compute $\boldsymbol{w}(s, a), \widetilde{\boldsymbol{w}}(s, a), \widehat{\boldsymbol{\sigma}}(s, a), \boldsymbol{Q}^{(0)}(s, a)$ without storing the samples. The updates can be computed in space $O(|\mathcal{S}||\mathcal{A}|)$ as well.

# 5  Comparison to Previous Work

| Algorithm | Sample Complexity | References |
|---|---|---|
| Phased Q-Learning | $\widetilde{O}(C\frac{|\mathcal{S}||\mathcal{A}|}{(1-\gamma)^7\epsilon^2})$ | [KS99] |
| Empirical QVI | $\widetilde{O}(\frac{|\mathcal{S}||\mathcal{A}|}{(1-\gamma)^5\epsilon^2})$ [4] | [AMK13] |
| Empirical QVI | $\widetilde{O}(\frac{|\mathcal{S}||\mathcal{A}|}{(1-\gamma)^3\epsilon^2})$ if $\epsilon = \widetilde{O}(\frac{1}{\sqrt{(1-\gamma)|\mathcal{S}|}})$ | [AMK13] |
| Randomized Primal-Dual Method | $\widetilde{O}(C\frac{|\mathcal{S}||\mathcal{A}|}{(1-\gamma)^4\epsilon^2})$ | [Wan17] |
| Sublinear Randomized Value Iteration | $\widetilde{O}\left(\frac{|\mathcal{S}||\mathcal{A}|}{(1-\gamma)^4\epsilon^2}\right)$ | [SWWY18] |
| Sublinear Randomized QVI | $\widetilde{O}\left(\frac{|\mathcal{S}||\mathcal{A}|}{(1-\gamma)^3\epsilon^2}\right)$ | This Paper |

Table 1: **Sample Complexity to Compute $\epsilon$-Approximate Policies Using the Generative Sampling Model**: Here $|\mathcal{S}|$ is the number of states, $|\mathcal{A}|$ is the number of actions per state, $\gamma \in (0, 1)$ is the discount factor, and $C$ is an upper bound on the ergodicity. Rewards are bounded between 0 and 1.

There exists a large body of literature on MDPs and RL (see e.g. [Kak03, SLL09, KBJ14, DB15] and reference therein). The classical MDP problem is to compute an optimal policy exactly or approximately, when the full MDP model is given as input. For a survey on existing complexity results when the full MDP model is given, see Appendix A.

Despite the aforementioned results of [Kak03, AMK13, SWWY18], there exists only a handful of additional RL methods that achieve a small sample complexity and a small run-time complexity at the same time for computing an $\epsilon$-optimal policy. A classical result is the phased Q-learning method by [KS99], which takes samples from the generative model and runs a randomized value iteration. The phased Q-learning method finds an $\epsilon$-optimal policy using $\mathcal{O}(|\mathcal{S}||\mathcal{A}|\epsilon^{-2}/\text{poly}(1-\gamma))$ samples/updates, where each update uses $\widetilde{O}(1)$ run time.[5] Another work [Wan17] gave a randomized mirror-prox method that applies to a special Bellman saddle point formulation of the DMDP. They achieve a total runtime of $\widetilde{O}(|\mathcal{S}|^3|\mathcal{A}|\epsilon^{-2}(1-\gamma)^{-6})$ for the general DMDP and $\widetilde{O}(C|\mathcal{S}||\mathcal{A}|\epsilon^{-2}(1-\gamma)^{-4})$ for DMDPs that are ergodic under all possible policies, where $C$ is a problem-specific ergodicity measure. A recent closely related work is [SWWY18] which gave a variance-reduced randomized value iteration that works with the generative model and finds an $\epsilon$-approximate policy in sample size/run time $\widetilde{O}(|\mathcal{S}||\mathcal{A}|\epsilon^{-2}(1-\gamma)^{-4})$, without requiring any ergodicity assumption.

Finally, in the case where $\epsilon = O\left(1/\sqrt{(1-\gamma)^{-1}|\mathcal{S}|}\right)$, [AMK13] showed that the solution obtained by performing exact PI on the empirical MDP model provides not only an $\epsilon$-optimal value but also an $\epsilon$-optimal policy. In this case, the number of samples is $\widetilde{O}(|\mathcal{S}||\mathcal{A}|(1-\gamma)^{-3}\epsilon^{-2})$ and matches the sample complexity lower bound. Although this sample complexity is optimal, it requires solving the empirical MDP exactly (see Appendix B), and is no longer sublinear in the size of the MDP model because of the very small approximation error $\epsilon = O(1/\sqrt{(1-\gamma)|\mathcal{S}|})$. See Table 1 for a list of comparable sample complexity results for solving MDP based on the generative model.

## 6    Concluding Remark

In summary, for a discounted Markov Decision Process (DMDP) $\mathcal{M} = (\mathcal{S}, \mathcal{A}, \boldsymbol{P}, \boldsymbol{r}, \gamma)$ provided we can only access the transition function of the DMDP through a generative sampling model, we provide an algorithm which computes an $\epsilon$-approximate policy with probability $1-\delta$ where both the time spent and number of sample taken is upper bounded by $\widetilde{O}((1-\gamma)^{-3}\epsilon^{-2}|\mathcal{S}||\mathcal{A}|)$. This improves upon the previous best known bounds by a factor of $1/(1-\gamma)$ and matches the the lower bounds proved in [AMK13] up to logarithmic factors.

The appendix is structured as follows. Section A surveys the existing runtime results for solving the DMDP when a full model is given. Section B provides an runtime optimal algorithm for computing approximate value functions (by directly combining [AMK13] and [SWWY18]). Section C gives technical analysis and variance upper bounds for the total-variance technique. Section D discusses sample complexity lower bounds for obtaining approximate policies with a generative sampling model. Section E provides proofs to lemmas, propositions and theorems in the main text of the paper. Section F extends our method and results to the finite-horizon MDP and provides a nearly matching sample complexity lower bound.

## Footnotes

[1]If instead the oracle needed time $\tau$, every running time result in this paper should be multiplied by $\tau$.

[2][AMK13] showed that one can obtain $\epsilon$-optimal *value* $v$ (instead of $\epsilon$-optimal policy) using sample size $\propto (1 - \gamma)^{-3}\epsilon^{-2}$. By using this $\epsilon$-optimal value $v$, one can get a greedy policy that is $[(1 - \gamma)^{-1}\epsilon]$-optimal. By setting $\epsilon \to (1 - \gamma)\epsilon$, one can obtain an $\epsilon$-optimal policy, using the number of samples $\propto (1 - \gamma)^{-5}\epsilon^{-2}$.

[3]A general $\boldsymbol{r} \in \mathbb{R}^{\mathcal{S} \times \mathcal{A}}$ can always be reduced to this case by shifting and scaling.

[4]Although not explicitly stated, an immediate derivation shows that obtaining an $\epsilon$-optimal policy in [AMK13] requires $O(|S||A|(1-\gamma)^{-5}\epsilon^{-2})$ samples.

[5]The dependence on $(1 - \gamma)$ in [KS99] is not stated explicitly but we believe basic calculations yield $O(1/(1-\gamma)^7)$.

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
