[Supplementary Material · nips2018_mdp_appendix_1.pdf]

# A   Previous Work on Solving DMDP with a Full Model

Value iteration was proposed by [Bel57] to compute an exact optimal policy of a given DMDP in time $\mathcal{O}((1-\gamma)^{-1}|\mathcal{S}|^2|\mathcal{A}|L\log((1-\gamma)^{-1}))$, where $L$ is the total number of bits needed to represent the input; and it can find an approximate $\epsilon$-approximate solution in time $\mathcal{O}(|\mathcal{S}|^2|\mathcal{A}|(1-\gamma)^{-1}\log(1/\epsilon(1-\gamma)))$; see e.g. [Tse90, LDK95]. The policy iteration was introduced by [How60] shortly after, where the policy is monotonically improved according to its associated value function. Its complexity has also been analyzed extensively; see e.g. [MS99, Ye11, Sch13]. Ye [Ye11] showed that policy iteration and the simplex method are strongly polynomial for DMDP and terminates in $\mathcal{O}(|\mathcal{S}|^2|\mathcal{A}|(1-\gamma)^{-1}\log(|\mathcal{S}|(1-\gamma)^{-1}))$ number of iterations. Later [HMZ13] and [Sch13] improved the iteration bound to $O(|\mathcal{S}||\mathcal{A}|(1-\gamma)^{-1}\log((1-\gamma)^1))$ for Howard's policy iteration method. A third approach is to formulate the nonlinear Bellman equation into a linear program [d'E63, DG60], and solve it using standard linear program solvers, such as the simplex method by Dantzig [Dan16] and the combinatorial interior-point algorithm by [Ye05]. [LS14, LS15] showed that one can solve linear programs in $\widetilde{O}(\sqrt{\mathrm{rank}(A)})$ number of linear system solves, which, applied to DMDP, yields to a running time of $\widetilde{O}(|\mathcal{S}|^{2.5}|\mathcal{A}|L)$ for computing the exact policy and $\widetilde{O}(|\mathcal{S}|^{2.5}|\mathcal{A}|\log(1/\epsilon))$ for computing an $\epsilon$-optimal policy. [SWWY18] further improved the complexity of value iteration by using randomization and variance reduction. See Table 2 for comparable run-time results or computing the optimal policy when the MDP model is fully given.

| Algorithm | Complexity | References |
|---|---|---|
| Value Iteration (exact) | $|\mathcal{S}|^2|\mathcal{A}|L\frac{\log(1/(1-\gamma))}{1-\gamma}$ | [Tse90, LDK95] |
| Value Iteration | $|\mathcal{S}|^2|\mathcal{A}|\frac{\log(1/(1-\gamma)\epsilon)}{1-\gamma}$ | [Tse90, LDK95] |
| Policy Iteration (Block Simplex) | $\frac{|\mathcal{S}|^4|\mathcal{A}|^2}{1-\gamma}\log(\frac{1}{1-\gamma})$ | [Ye11],[Sch13] |
| Recent Interior Point Methods | $\widetilde{O}(|\mathcal{S}|^{2.5}|\mathcal{A}|L)\ \widetilde{O}(|\mathcal{S}|^{2.5}|\mathcal{A}|\log(1/\epsilon))$ | [LS14] |
| Combinatorial Interior Point Algorithm | $|\mathcal{S}|^4|\mathcal{A}|^4\log\frac{|\mathcal{S}|}{1-\gamma}$ | [Ye05] |
| High Precision Randomized Value Iteration | $\widetilde{O}\left[\left(\mathrm{nnz}(P)+\frac{|\mathcal{S}||\mathcal{A}|}{(1-\gamma)^3}\right)\log\left(\frac{1}{\epsilon\delta}\right)\right]$ | [SWWY18] |

Table 2:  **Running Times to Solve DMDPs Given the Full MDP Model**: In this table, $|\mathcal{S}|$ is the number of states, $|\mathcal{A}|$ is the number of actions per state, $\gamma \in (0,1)$ is the discount factor, and $L$ is a complexity measure of the linear program formulation that is at most the total bit size to present the DMDP input. Rewards are bounded between 0 and 1.

# B   Sample and Time Efficient Value Computation

In this section, we describe an algorithm that obtains an $\epsilon$-optimal values in time $\widetilde{O}(\epsilon^{-2}(1-\gamma)^{-3}|\mathcal{S}||\mathcal{A}|)$. Note that the time and number of samples of this algorithm is optimal (up to logarithmic factors) due to the lower bound in [AMK13] which also established this upper bound on the sample complexity (but not time complexity) of the problem.

We achieve this by combining the algorithms in [AMK13] and [SWWY18]. First, we use the ideas and analysis of [AMK13] to construct a sparse MDP where the optimal value function of this MDP approximates the optimal value function of the original MDP and then we run the high precision algorithm in [SWWY18] on this sparsified MDP. We show that [SWWY18] runs in nearly linear time on sparsified MDP. Since the number of samples taken to construct the sparsified MDP was the the optimal number of samples, to solve the problem, the ultimate running time we thereby achieve is nearly optimal as any algorithm needs spend time at least the number of samples to obtain these samples.

We include this for completeness but note that the approximate value function we show how to compute here does not suffice to compute policy of the MDP of comparable quality. The greedy policy of an $\epsilon$-optimal value function is an $\epsilon/(1-\gamma)$-optimal policy in the worst case. It has been shown in [AMK13] that the greedy policy of their value function is $\epsilon$-optimal if $\epsilon \leq (1-\gamma)^{1/2}|\mathcal{S}|^{-1/2}$. However, when $\epsilon$ is so small, the seemingly sublinear runtime $\widetilde{O}((1-\gamma)^{-3}\mathcal{S}||\mathcal{A}|/\epsilon^2)$ essentially

means a linear running time and sample complexity as $O((1 - \gamma)^{-3}|\mathcal{S}|^2|\mathcal{A}|)$. The running time can be obtained by merely applying the result in [SWWY18] (although with a slightly different computation model).

## B.1 The Sparsified DMDP

Suppose we are given a DMDP $\mathcal{M} = (\mathcal{S}, \mathcal{A}, \boldsymbol{r}, \boldsymbol{P}, \gamma)$ with a sampling oracle. To approximate the optimal value of this MDP, we perform a spasification procedure as in [AMK13]. Sparsification of DMDP is conducted as follows. Let $\delta > 0, \epsilon > 0$ be arbitrary. First we pick a number

$$m = \Theta \left[ \frac{1}{(1 - \gamma)^3 \epsilon^2} \log \left( \frac{|\mathcal{S}||\mathcal{A}|}{\delta} \right) \right]. \tag{B.1}$$

For each $s \in \mathcal{S}$ and each $a \in \mathcal{A}$, we generate a sequence of independent samples from $\mathcal{S}$ using the probability vector $\boldsymbol{P}_{s,a}$

$$s_{s,a}^{(1)}, s_{s,a}^{(2)}, \ldots, s_{s,a}^{(m)}.$$

Next we construct a new and sparse probability vector $\widehat{\boldsymbol{P}}_{s,a} \in \Delta_{|\mathcal{S}|}$ as

$$\forall s' \in \mathcal{S} : \widehat{\boldsymbol{P}}_{s,a}(s') = \frac{1}{m} \cdot \sum_{i=1}^{m} \mathbf{1}(s_{s,a}^{(i)} = s').$$

Combining these $|\mathcal{S}||\mathcal{A}|$ new probability vectors, we obtain a new probability transition matrix $\widehat{\boldsymbol{P}} \in \mathbb{R}^{\mathcal{S} \times \mathcal{A} \times \mathcal{S}}$ with number of non-zeros

$$\mathrm{nnz}(\widehat{\boldsymbol{P}}) = O \left[ \frac{|\mathcal{S}||\mathcal{A}|}{(1 - \gamma)^3 \epsilon^2} \log \left( \frac{|\mathcal{S}||\mathcal{A}|}{\delta} \right) \right].$$

Denote $\widehat{\mathcal{M}} = (\mathcal{S}, \mathcal{A}, \boldsymbol{r}, \widehat{\boldsymbol{P}}, \gamma)$ as the sparsified DMDP. In the rest of this section, we use $\widehat{\cdot}$ to represent the quantities corresponding to DMDP $\widehat{\mathcal{M}}$, e.g., $\widehat{\boldsymbol{v}}^*$ for the optimal value function, $\widehat{\pi}^*$ for a optimal policy, and $\widehat{\boldsymbol{Q}}^*$ for the optimal $Q$-function. There is a strong approximation guarantee of the optimal $Q$-function of the sparsified MDP, presented as follows.

**Theorem B.1** ([AMK13]). *Let $\mathcal{M}$ be the original DMDP and $\widehat{\mathcal{M}}$ be the corresponding sparsified version. Let $\boldsymbol{Q}^*$ be the optimal Q-function vector of the original DMDP and $\widehat{\boldsymbol{Q}}^*$ be the optimal Q-function of $\widehat{\mathcal{M}}$. Then with probability at least $1 - \delta$ (over the randomness of the samples),*

$$\|\widehat{\boldsymbol{Q}}^* - \boldsymbol{Q}^*\|_\infty \leq \epsilon.$$

Recall that $\boldsymbol{v}^*$ and $\widehat{\boldsymbol{v}}^*$ are the optimal value functions of $\mathcal{M}$ and $\widehat{\mathcal{M}}$. From Theorem B.1, we immediately have

$$\forall s \in \mathcal{S} : |\boldsymbol{v}^*(s) - \widehat{\boldsymbol{v}}^*(s)| = |\max_{a \in \mathcal{A}} \boldsymbol{Q}^*(s, a) - \max_{a \in \mathcal{A}} \widehat{\boldsymbol{Q}}^*(s, a)| \leq \max_{a \in \mathcal{A}} |\boldsymbol{Q}^*(s, a) - \widehat{\boldsymbol{Q}}^*(s, a)| \leq \epsilon,$$

with probability at least $1 - \delta$.

## B.2 High Precision Algorithm in the Sparsified MDP

Next we shall use the high precision algorithm of the [SWWY18] which has the following guarantee.

**Theorem B.2** ([SWWY18]). *There is an algorithm which given an input DMDP $\mathcal{M} = (\mathcal{S}, \mathcal{A}, \boldsymbol{r}, \boldsymbol{P}, \gamma)$ in time[6]*

$$\widetilde{O} \left[ \left( \mathrm{nnz}(\boldsymbol{P}) + \frac{|\mathcal{S}||\mathcal{A}|}{(1 - \gamma)^3} \right) \cdot \log \epsilon^{-1} \cdot \log \delta^{-1} \right]$$

*and outputs a vector $\widetilde{\boldsymbol{v}}^*$ such that with probability at least $1 - \delta$,*

$$\|\widetilde{\boldsymbol{v}}^* - \boldsymbol{v}^*\|_\infty \leq \epsilon.$$

*where $\boldsymbol{v}^*$ is the optimal value of $\mathcal{M}$.*

Combining the above two theorems, we immediately obtain an algorithm for finding $\epsilon$-optimal value functions. It works by first generating enough samples for each state-action pair and then call the high-precision MDP solver by [SWWY18]. It does not sample transitions adaptively. We show that it achieves an optimal running time guarantee (up to $\mathrm{poly}\log$ factors) of obtaining the value function under the sampling oracle model.

**Theorem B.3.** *Given an input DMDP $\mathcal{M} = (\mathcal{S}, \mathcal{A}, \boldsymbol{r}, \boldsymbol{P}, \gamma)$ with a sampling oracle and optimal value function $\boldsymbol{v}^*$, there exists an algorithm, that runs in time*

$$\widetilde{O}\left( \frac{|\mathcal{S}||\mathcal{A}|}{(1-\gamma)^3} \cdot \frac{1}{\epsilon^2} \cdot \log^2\left(\frac{1}{\delta}\right) \right)$$

*and outputs a vector $\widehat{\boldsymbol{v}}^*$ such that $\|\widehat{\boldsymbol{v}}^* - \boldsymbol{v}^*\|_\infty \leq O(\epsilon)$ with probability at least $1 - O(\delta)$.*

*Proof.* We first obtain a sparsified MDP $\widehat{\mathcal{M}} = (\mathcal{S}, \mathcal{A}, \boldsymbol{r}, \widehat{\boldsymbol{P}}, \gamma)$ using the procedure described in Section B.1. This procedure runs in time $O(|\mathcal{S}||\mathcal{A}|m)$, recalling that $m$ is the number of samples per $(s, a)$, defined in (B.1). Let $\widehat{\boldsymbol{u}}^*$ be the optimal value function of $\widehat{\mathcal{M}}$. By Theorem B.1, with probability at least $1 - \delta$, $\|\widehat{\boldsymbol{u}}^* - \boldsymbol{v}^*\| \leq \epsilon$, which we condition on for the rest of the proof. Calling the algorithm in Theorem B.2, we obtain a vector $\widetilde{\boldsymbol{u}}^*$ in time

$$\widetilde{O}\left[ \left( \mathrm{nnz}(\widehat{\boldsymbol{P}}) + \frac{|\mathcal{S}||\mathcal{A}|}{(1-\gamma)^3} \right) \cdot \log \epsilon^{-1} \cdot \log \delta^{-1} \right] = \widetilde{O}\left( \frac{|\mathcal{S}||\mathcal{A}|}{(1-\gamma)^3} \cdot \frac{1}{\epsilon^2} \cdot \log^2 \frac{1}{\delta} \right)$$

and that with probability at least $1-\delta$, $\|\widetilde{\boldsymbol{u}}^* - \widehat{\boldsymbol{u}}^*\| \leq \epsilon$, which we condition on. By triangle inequality, we have

$$\|\widetilde{\boldsymbol{u}}^* - \boldsymbol{v}^*\|_\infty \leq \|\widetilde{\boldsymbol{u}}^* - \widehat{\boldsymbol{u}}^*\|_\infty + \|\widehat{\boldsymbol{u}}^* - \boldsymbol{v}^*\|_\infty \leq 2\epsilon.$$

This concludes the proof. $\qquad\square$

## C   Variance Bounds

In this section, we study some properties of a DMDP. Most of the content in this section is similar to [AMK13]. We provide slight modifications and improvement to make the results fit to our application. The main result of this section is to show the following lemma.

**Lemma C.1** (Upper Bound on Variance). *For any $\pi$, we have*

$$\left\| (\boldsymbol{I} - \gamma \boldsymbol{P}^\pi)^{-1} \sqrt{\boldsymbol{\sigma}_{\boldsymbol{v}^\pi}} \right\|_\infty^2 \leq \frac{1 + \gamma}{\gamma^2 (1 - \gamma)^3},$$

*where $\sigma_{v^\pi} = \boldsymbol{P}^\pi (\boldsymbol{v}^\pi)^2 - (\boldsymbol{P}^\pi \boldsymbol{v}^\pi)^2$ is the "one-step" variance of playing policy $\pi$.*

Before we prove this lemma, we introduce another notation. We define $\boldsymbol{\Sigma}^\pi \in \mathbb{R}^{|\mathcal{S}||\mathcal{A}|}$ for all $(s, a) \in \mathcal{S} \times \mathcal{A}$ by

$$\boldsymbol{\Sigma}^\pi(s, a) := \mathbb{E}\left[ \left( \boldsymbol{r}(s, a) + \sum_{t \geq 1} \gamma^t \boldsymbol{r}(s^t, a^t) - \boldsymbol{Q}^\pi(s, a) \right)^2 \middle| s^0 = s, a^0 = a, a^t = \pi(s^t) \right]$$

where $a^t = \pi(s^t)$. Thus $\boldsymbol{\Sigma}^\pi$ is the variance of the reward of starting with $(s, a)$ and play $\pi$ for infinite steps. The crucial observation of obtaining the near-optimal sample complexity is the following "Bellman Equation" for variance. It is a consequence of "the law of total variance".

**Lemma C.2** (Bellman Equation for variance). *$\boldsymbol{\Sigma}^\pi$ satisfies the Bellman equation*

$$\boldsymbol{\Sigma}^\pi = \gamma^2 \boldsymbol{\sigma}_{\boldsymbol{v}^\pi} + \gamma^2 \cdot \boldsymbol{P}^\pi \boldsymbol{\Sigma}^\pi.$$

*Proof.* By direct expansion,

$$\boldsymbol{\Sigma}^\pi(s, a) = \mathbb{E}\left[ \left( \boldsymbol{r}(s, a) + \sum_{t \geq 1} \gamma^t \boldsymbol{r}(s^t, a^t) \right)^2 \middle| s^0 = s, a^0 = a, a^t = \pi(s^t) \right] - (\boldsymbol{Q}^\pi(s, a))^2. \quad \text{(C.1)}$$

The first term in RHS can be written as

$$\mathbb{E}\left[\left(\boldsymbol{r}(s,a) + \sum_{t\geq 1}\gamma^t \boldsymbol{r}(s^t,a^t)\right)^2 \Bigg| s^0 = s, a^0 = a, a^t = \pi(s^t)\right]$$

$$= \sum_{s'\in\mathcal{S}}\boldsymbol{P}_{s,a}(s')\mathbb{E}\left[\left(\boldsymbol{r}(s,a) + \gamma\boldsymbol{r}(s',\pi(s')) + \gamma\sum_{t\geq 1}\gamma^t \boldsymbol{r}(s^t,a^t)\right)^2 \Bigg| s^0 = s', a^0 = \pi(s'), a^t = \pi(s^t)\right]$$

$$= \boldsymbol{r}(s,a)^2 + 2\gamma\boldsymbol{r}(s,a)\cdot\sum_{s'\in\mathcal{S}}\boldsymbol{P}_{s,a}(s')\boldsymbol{Q}^\pi(s',\pi(s'))$$

$$\quad + \gamma^2\sum_{s'\in\mathcal{S}}\boldsymbol{P}_{s,a}(s')\mathbb{E}\left[\left(\boldsymbol{r}(s',\pi(s')) + \sum_{t\geq 1}\gamma^t \boldsymbol{r}(s^t,a^t)\right)^2 \Bigg| s^0 = s', a^0 = \pi(s'), a^t = \pi(s^t)\right]$$

$$= \boldsymbol{r}(s,a)^2 + 2\gamma\boldsymbol{r}(s,a)\cdot\sum_{s'\in\mathcal{S}}\boldsymbol{P}_{s,a}(s')\boldsymbol{Q}^\pi(s',\pi(s')) + \gamma^2(\boldsymbol{P}^\pi\boldsymbol{\Sigma}^\pi)(s,a) + \gamma^2\sum_{s'\in\mathcal{S}}\boldsymbol{P}_{s,a}(s')(\boldsymbol{Q}^\pi(s',\pi(s')))^2$$

$$= \boldsymbol{Q}^\pi(s,a)^2 + \gamma^2(\boldsymbol{P}^\pi\boldsymbol{\Sigma}^\pi)(s,a) + \gamma^2\sum_{s'\in\mathcal{S}}\boldsymbol{P}_{s,a}(s')(\boldsymbol{Q}^\pi(s',\pi(s')))^2 - \gamma^2\left(\sum_{s'\in\mathcal{S}}\boldsymbol{P}_{s,a}(s')\boldsymbol{Q}^\pi(s',\pi(s'))\right)^2$$

$$= \boldsymbol{Q}^\pi(s,a)^2 + \gamma^2(\boldsymbol{P}^\pi\boldsymbol{\Sigma}^\pi)(s,a) + \gamma^2\boldsymbol{\sigma}_{\boldsymbol{v}^\pi}(s,a).$$

Combining the above two equations, we conclude the proof. $\qquad\square$

As a remark, we note that

$$\boldsymbol{\Sigma}^\pi = \gamma^2(\boldsymbol{I} - \gamma^2\boldsymbol{P}^\pi)^{-1}\boldsymbol{\sigma}_{\boldsymbol{v}^\pi}.$$

Furthermore, by definition, we have

$$\max_{(s,a)\in\mathcal{S}}\boldsymbol{\Sigma}^\pi(s,a) \leq (1-\gamma)^{-2},$$

The next lemma is crucial in proving the error bounds.

**Lemma C.3.** *Let $\boldsymbol{P} \in \mathbb{R}^{n\times n}$ be a non-negative matrix in which every row has $\ell_1$ norm at most 1, i.e. $\ell_\infty$ operator norm at most 1. Then for all $\gamma \in (0,1)$ and $\boldsymbol{v} \in \mathbb{R}^n_{\geq 0}$ we have*

$$\|(\boldsymbol{I} - \gamma\boldsymbol{P})^{-1}\sqrt{\boldsymbol{v}}\|_\infty \leq \sqrt{\frac{1}{1-\gamma}\|(\boldsymbol{I} - \gamma\boldsymbol{P})^{-1}\boldsymbol{v}\|_\infty} \leq \sqrt{\frac{1+\gamma}{1-\gamma}\|(\boldsymbol{I} - \gamma^2\boldsymbol{P})^{-1}\boldsymbol{v}\|_\infty}.$$

*Proof.* Since, every row of $\boldsymbol{P}$ has $\ell_1$ norm at most 1, by Cauchy-Schwarz for $i \in [n]$ we have

$$[\boldsymbol{P}\sqrt{\boldsymbol{v}}]_i = \sum_{j\in[n]}\boldsymbol{P}_{ij}\sqrt{\boldsymbol{v}}_j \leq \sqrt{\sum_{j\in[n]}\boldsymbol{P}_{ij}\cdot\sum_{j\in[n]}\boldsymbol{P}_{ij}\boldsymbol{v}_j} \leq \sqrt{\boldsymbol{P}\boldsymbol{v}}.$$

Since $\boldsymbol{v}$ is non-negative and applying $\boldsymbol{P}$ preserves non-negativity, applying this inequality repeatedly yields that $\boldsymbol{P}^k\sqrt{\boldsymbol{v}} \leq \sqrt{\boldsymbol{P}^k\boldsymbol{v}}$ entrywise for all $k > 0$. Consequently, Cauchy-Schwarz again yields

$$(\boldsymbol{I} - \gamma\boldsymbol{P})^{-1}\sqrt{\boldsymbol{v}} = \sum_{i=0}^\infty[\gamma\boldsymbol{P}]^i\sqrt{\boldsymbol{v}} \leq \sum_{i=0}^\infty\gamma^i\sqrt{\boldsymbol{P}^i\boldsymbol{v}} \leq \sqrt{\sum_{i=0}^\infty\gamma^i\cdot\sum_{i=0}^\infty\gamma^i\boldsymbol{P}^i v}$$

$$\leq \sqrt{\frac{1}{1-\gamma}\|(\boldsymbol{I} - \gamma\boldsymbol{P})^{-1}\boldsymbol{v}\|_\infty}.$$

Next, as $(\boldsymbol{I} - \gamma\boldsymbol{P})(\boldsymbol{I} + \gamma\boldsymbol{P}) = (\boldsymbol{I} - \gamma\boldsymbol{P}^2)$ we see that $(\boldsymbol{I} - \gamma\boldsymbol{P})^{-1} = (\boldsymbol{I} + \gamma\boldsymbol{P})(\boldsymbol{I} - \gamma^2\boldsymbol{P})^{-1}$. Furthermore, as $\|\boldsymbol{P}x\|_\infty \leq \|x\|_\infty$ for all $\boldsymbol{x}$ we have $\|(\boldsymbol{I} + \gamma\boldsymbol{P})x\|_\infty \leq (1+\gamma)\|x\|_\infty$ for all $\boldsymbol{x}$ and therefore $\|(\boldsymbol{I} - \gamma\boldsymbol{P})^{-1}\boldsymbol{v}\|_\infty \leq (1+\gamma)\|(\boldsymbol{I} - \gamma^2\boldsymbol{P})^{-1}\boldsymbol{v}\|_\infty$ as desired. $\qquad\square$

We are now ready to prove Lemma C.1.

*Proof of Lemma C.1.* The lemma follows directly from the application of Lemma C.3. This proof is slightly simpler, tighter, and more general than the one in [AMK13]. $\qquad\square$

# D Lower Bounds on Policy

**Lemma D.1.** *Suppose $\mathcal{M} = (\mathcal{S}, \mathcal{A}, P, \gamma, \boldsymbol{r})$ is a DMDP with an sampling oracle. Suppose $\pi$ is a given policy. Then there is an algorithm, halts in $\widetilde{O}((1-\gamma)^{-3}\epsilon^{-2}|\mathcal{S}|)$ time, outputs a vector $\boldsymbol{v}$ such that, with high probability, $\|\boldsymbol{v}^\pi - \boldsymbol{v}\|_\infty \leq \epsilon$.*

*Proof.* The lemma follows from a direct application of Theorem B.2. $\qquad\square$

*Remark* D.2. Suppose $|\mathcal{A}| = \widetilde{\Omega}(1)$. Suppose there is an algorithm that obtains an $\epsilon$-optimal policy with $Z$ samples, then the above lemma implies an algorithm for obtaining an $\epsilon$-optimal value function with $Z + \widetilde{O}((1-\gamma)^{-3}\epsilon^{-2}|\mathcal{S}|)$ samples. By the $\Omega((1-\gamma)^{-3}\epsilon^{-2}|\mathcal{S}||\mathcal{A}|)$ sample bound on obtaining approximate value functions given in [AMK13], the above lemma implies a

$$Z = \Omega((1-\gamma)^{-3}\epsilon^{-2}|\mathcal{S}||\mathcal{A}|) - \widetilde{O}((1-\gamma)^{-3}\epsilon^{-2}|\mathcal{S}|) = \Omega((1-\gamma)^{-3}\epsilon^{-2}|\mathcal{S}||\mathcal{A}|)$$

sample lower bound for obtaining an $\epsilon$-optimal policy.

# E Missing Proofs

Here are several standard properties of the Bellman value operator (see, e.g., [Ber13]).

**Fact 1.** *Let $\boldsymbol{v}_1, \boldsymbol{v}_2 \in \mathbb{R}^\mathcal{S}$ be two vectors. Let $\mathcal{T}$ be a value operator of a DMDP with discount factor $\gamma$. Let $\pi \in \mathcal{A}^\mathcal{S}$ be an arbitrary policy. Then the follows hold.*

- ***Monotonicity***: *If $\boldsymbol{v}_1 \leq \boldsymbol{v}_2$ then $\mathcal{T}(\boldsymbol{v}_1) \leq \mathcal{T}(\boldsymbol{v}_2)$;*
- ***Contraction***: *$\|\mathcal{T}(\boldsymbol{v}_1) - \mathcal{T}(\boldsymbol{v}_2)\|_\infty \leq \gamma\|\boldsymbol{v}_1 - \boldsymbol{v}_2\|_\infty$ and $\|\mathcal{T}_\pi(\boldsymbol{v}_1) - \mathcal{T}_\pi(\boldsymbol{v}_2)\|_\infty \leq \gamma\|\boldsymbol{v}_1 - \boldsymbol{v}_2\|_\infty$.*

## E.1 Missing Proofs from Section 4

To begin, we introduce two standard concentration results. Let $\boldsymbol{p} \in \Delta_\mathcal{S}$ be a probability vector, and $\boldsymbol{v} \in \mathbb{R}^\mathcal{S}$ be a vector. Let $\boldsymbol{p}_m \in \Delta_\mathcal{S}$ be empirical estimations of $\boldsymbol{p}$ using $m$ i.i.d. samples from the distribution $\boldsymbol{p}$. For instance, let these samples be $s_1, s_2, \ldots, s_m \in \mathcal{S}$, then $\forall s \in \mathcal{S} : \boldsymbol{p}_m(s) = \sum_{j=1}^m \mathbf{1}(s_j = s)/m$.

**Theorem E.1** (Hoeffding Inequality). *Let $\delta \in (0, 1)$ be a parameter, vectors $\boldsymbol{p}, \boldsymbol{p}_m$ and $\boldsymbol{v}$ defined above. Then with probability at least $1 - \delta$,*

$$\left|\boldsymbol{p}^\top \boldsymbol{v} - \boldsymbol{p}_m^\top \boldsymbol{v}\right| \leq \|\boldsymbol{v}\|_\infty \cdot \sqrt{2m^{-1}\log(2\delta^{-1})}.$$

**Theorem E.2** (Bernstein Inequality). *Let $\delta \in (0, 1)$ be a parameter, vectors $\boldsymbol{p}, \boldsymbol{p}_m$ and $\boldsymbol{v}$ defined as in Theorem E.1. Then with probability at least $1 - \delta$*

$$\left|\boldsymbol{p}^\top \boldsymbol{v} - \boldsymbol{p}_m^\top \boldsymbol{v}\right| \leq \sqrt{2m^{-1}\underset{s' \sim \boldsymbol{p}}{\mathrm{Var}}(\boldsymbol{v}(s')) \cdot \log(2\delta^{-1})} + (2/3)m^{-1}\|\boldsymbol{v}\|_\infty \cdot \log(2\delta^{-1}),$$

*where $\underset{s' \sim \boldsymbol{p}}{\mathrm{Var}}(\boldsymbol{v}(s')) = \boldsymbol{p}^\top \boldsymbol{v}^2 - (\boldsymbol{p}^\top \boldsymbol{v})^2$.*

*Proof of Lemma 4.1.* By Theorem E.2 and a union bound over all $(s, a)$ pairs, with probability at least $1 - \delta/4$, for every $(s, a)$, we have

$$\left|\widetilde{\boldsymbol{w}}(s, a) - \boldsymbol{P}_{s,a}^\top \boldsymbol{v}^{(0)}\right| \leq \sqrt{2\sigma_{\boldsymbol{v}^{(0)}} \cdot m_1^{-1} \cdot L} + 2 \cdot (3m_1)^{-1} \cdot \|\boldsymbol{v}^{(0)}\|_\infty \cdot L, \tag{E.1}$$

which is the first inequality.

Next, by Theorem E.1 and a union bound over all $(s, a)$ pairs, with probability at least $1 - \delta/4$, for every $(s, a)$, we have

$$\left|\widetilde{\boldsymbol{w}}(s, a) - \boldsymbol{P}_{s,a}^\top \boldsymbol{v}^{(0)}\right| \leq \|\boldsymbol{v}^{(0)}\|_\infty \cdot \sqrt{2m_1^{-1}L},$$

which we condition on. Thus

$$\left|\widetilde{\boldsymbol{w}}(s,a)^2 - (\boldsymbol{P}_{s,a}^\top \boldsymbol{v}^{(0)})^2\right| = (\widetilde{\boldsymbol{w}}(s,a) + \boldsymbol{P}_{s,a}^\top \boldsymbol{v}^{(0)}) \cdot |\widetilde{\boldsymbol{w}}(s,a) - \boldsymbol{P}_{s,a}^\top \boldsymbol{v}^{(0)}|$$

$$\leq \left[2\boldsymbol{P}_{s,a}^\top \boldsymbol{v}^{(0)} + \|\boldsymbol{v}^{(0)}\|_\infty \cdot \sqrt{2m_1^{-1}L}\right] \cdot |\widetilde{\boldsymbol{w}}(s,a) - \boldsymbol{P}_{s,a}^\top \boldsymbol{v}^{(0)}|$$

$$\leq 2(\boldsymbol{P}_{s,a}^\top \boldsymbol{v}^{(0)}) \cdot \|\boldsymbol{v}^{(0)}\|_\infty \cdot \sqrt{2m_1^{-1}L} + \|\boldsymbol{v}^{(0)}\|_\infty^2 \cdot 2m_1^{-1}L.$$

Since $\boldsymbol{P}_{s,a}^\top \boldsymbol{v}^{(0)} \leq \|\boldsymbol{v}^{(0)}\|_\infty$, we obtain

$$\left|\widetilde{\boldsymbol{w}}(s,a)^2 - (\boldsymbol{P}_{s,a}^\top \boldsymbol{v}^{(0)})^2\right| \leq 3\|\boldsymbol{v}^{(0)}\|_\infty^2 \cdot \sqrt{2m_1^{-1}L},$$

provided $2m_1^{-1}L \leq 1$. Next by Lemma E.1 and a union bound over all $(s,a)$ pairs, with probability at least $1 - \delta/4$, for every $(s,a)$, we have

$$\left|\frac{1}{m_1}\sum_{j=1}^{m_1} \boldsymbol{v}^2(s_{s,a}^{(j)}) - \boldsymbol{P}_{s,a}^\top \boldsymbol{v}^2\right| \leq \|\boldsymbol{v}^{(0)}\|_\infty^2 \cdot \sqrt{2L/m_1}.$$

By a union bound, we obtain, with probability at least $1 - \delta/2$,

$$\left|\widehat{\boldsymbol{\sigma}}(s,a) - \boldsymbol{\sigma}_{\boldsymbol{v}^{(0)}}(s,a)\right| \leq \left|\widetilde{\boldsymbol{w}}(s,a)^2 - (\boldsymbol{P}_{s,a}^\top \boldsymbol{v}^{(0)})^2\right| + \left|m_1^{-1}\sum_{j=1}^{m_1} \boldsymbol{v}^2(s_{s,a}^{(j)}) - \boldsymbol{P}_{s,a}^\top \boldsymbol{v}^2\right|$$

$$\leq 4\|\boldsymbol{v}^{(0)}\|_\infty^2 \cdot \sqrt{2m_1^{-1}L}. \tag{E.2}$$

By a union bound, with probability at least $1-\delta$, both (E.1) and (E.2) hold, concluding the proof. $\quad\square$

*Proof of Lemma 4.2.* Since for each $(s,a)$, $\boldsymbol{\sigma}_{\boldsymbol{v}}(s,a)$ is a variance, then we have triangle inequality,

$$\sqrt{\boldsymbol{\sigma}_{\boldsymbol{v}}} \leq \sqrt{\boldsymbol{\sigma}_{\boldsymbol{v}^*}} + \sqrt{\boldsymbol{\sigma}_{\boldsymbol{v}-\boldsymbol{v}^*}}.$$

Observing that

$$\boldsymbol{\sigma}_{\boldsymbol{v}-\boldsymbol{v}^*}(s,a) \leq \boldsymbol{P}_{s,a}^\top (\boldsymbol{v} - \boldsymbol{v}^*)^2 \leq \epsilon^2 \cdot \boldsymbol{1}.$$

We conclude the proof by taking a square root of all three sides of the above inequality. $\quad\square$

*Proof of Lemma 4.3.* Recall that for each $(s,a) \in \mathcal{S} \times \mathcal{A}$,

$$\boldsymbol{g}^{(i)}(s,a) = \frac{1}{m_2}\sum_{j=1}^{m_2} \left[\boldsymbol{v}^{(i)}(s_{s,a}^{(j)}) - \boldsymbol{v}^{(0)}(s_{s,a}^{(j)})\right] - (1-\gamma)\frac{u}{8},$$

where $m_2 = 128(1-\gamma)^{-2} \cdot \log(2|\mathcal{S}||\mathcal{A}|R/\delta)$ and $s_{s,a}^{(1)}, s_{s,a}^{(2)}, \ldots, s_{s,a}^{(m_2)}$ is a sequence of independent samples from $\boldsymbol{P}_{s,a}$. Thus by Theorem E.1 and a union bound over $\mathcal{S} \times \mathcal{A}$, with probability at least $1 - \delta/R$, we have

$$\forall (s,a) \in \mathcal{S} \times \mathcal{A}: \left|\sum_{j=1}^{m_2} \left[\boldsymbol{v}^{(i)}(s_{s,a}^{(j)}) - \boldsymbol{v}^{(0)}(s_{s,a}^{(j)})\right] - \boldsymbol{P}_{s,a}^\top \left[\boldsymbol{v}^{(i)} - \boldsymbol{v}^{(0)}\right]\right|$$

$$\leq \|\boldsymbol{v}^{(i)} - \boldsymbol{v}^{(0)}\|_\infty \sqrt{2m_2^{-1}\log(2|\mathcal{S}||\mathcal{A}|\delta'^{-1})} \leq (1-\gamma)u/8.$$

Finally by shifting the estimate to have one-sided error, we obtain the one-side error $(1-\gamma)u/4$ in the statement of this lemma. $\quad\square$

*Proof of Lemma 4.4.* For $i = 0$, $\boldsymbol{Q}^{(0)} = \boldsymbol{r} + \gamma\boldsymbol{w}$. By Lemma 4.1, with probability at least $1 - \delta$,

$$|\widetilde{\boldsymbol{w}} - \boldsymbol{P}\boldsymbol{v}^{(0)}| \leq \sqrt{2\alpha_1 \boldsymbol{\sigma}_{\boldsymbol{v}^{(0)}}} + \frac{2}{3} \cdot \alpha_1 \cdot \|\boldsymbol{v}^{(0)}\|_\infty \boldsymbol{1},$$

and

$$\left|\widehat{\boldsymbol{\sigma}} - \boldsymbol{\sigma}_{\boldsymbol{v}^{(0)}}\right| \leq 4\|\boldsymbol{v}^{(0)}\|_\infty^2 \cdot \sqrt{2\alpha_1}\boldsymbol{1}, \tag{E.3}$$

which we condition on. We have

$$|\widetilde{\boldsymbol{w}} - \boldsymbol{P}\boldsymbol{v}^{(0)}| \le \sqrt{2\alpha_1\widehat{\boldsymbol{\sigma}}} + (4\alpha_1^{3/4}\|\boldsymbol{v}^{(0)}\|_\infty + \frac{2}{3}\cdot\alpha_1\cdot\|\boldsymbol{v}^{(0)}\|_\infty)\mathbf{1}.$$

Thus

$$\boldsymbol{w} = \widetilde{\boldsymbol{w}} - \sqrt{2\alpha_1\widehat{\boldsymbol{\sigma}}} - 4\alpha_1^{3/4}\|\boldsymbol{v}^{(0)}\|_\infty\mathbf{1} - \frac{2}{3}\cdot\alpha_1\cdot\|\boldsymbol{v}^{(0)}\|_\infty\mathbf{1} \le \boldsymbol{P}\boldsymbol{v}^{(0)}, \tag{E.4}$$

and

$$\boldsymbol{w} \ge \boldsymbol{P}\boldsymbol{v}^{(0)} - 2\sqrt{2\alpha_1\widehat{\boldsymbol{\sigma}}} - (8\alpha_1^{3/4}\|\boldsymbol{v}^{(0)}\|_\infty + \frac{4}{3}\cdot\alpha_1\cdot\|\boldsymbol{v}^{(0)}\|_\infty)\mathbf{1}.$$

By (E.3) and Lemma 4.2, we have

$$\sqrt{\widehat{\boldsymbol{\sigma}}} \le \sqrt{\boldsymbol{\sigma}_{\boldsymbol{v}^{(0)}}} + 2\|\boldsymbol{v}^{(0)}\|_\infty(2\alpha)^{1/4}\mathbf{1} \le \sqrt{\boldsymbol{\sigma}_{\boldsymbol{v}^*}} + u\mathbf{1} + 2\|\boldsymbol{v}^{(0)}\|_\infty(2\alpha)^{1/4}\mathbf{1}.$$

we have

$$\boldsymbol{w} \ge \boldsymbol{P}\boldsymbol{v}^{(0)} - 2\sqrt{2\alpha_1\boldsymbol{\sigma}_{\boldsymbol{v}^*}} - 2\sqrt{2\alpha_1}u\mathbf{1} - 16\alpha_1^{3/4}\|\boldsymbol{v}^{(0)}\|_\infty\mathbf{1} - \frac{4}{3}\cdot\alpha_1\cdot\|\boldsymbol{v}^{(0)}\|_\infty\mathbf{1} \tag{E.5}$$

For the rest of the proof, we condition on the event that (E.4) and (E.5) hold, which happens with probability at least $1 - \delta$. Denote $\boldsymbol{v}^{(-1)} = \mathbf{0}$. Thus we have $\boldsymbol{v}^{(-1)} \le \boldsymbol{v}^{(0)} \le \mathcal{T}_{\pi^{(0)}}(\boldsymbol{v}^{(0)})$. Next we prove the lemma by induction on $i$. Assume for some $i \ge 1$, with probability at least $1 - (i-1)\delta'$ the following holds,

$$\forall 0 \le k \le i-1: \quad \boldsymbol{v}^{(k-1)} \le \boldsymbol{v}^{(k)} \le \mathcal{T}_{\pi^{(k)}}(\boldsymbol{v}^{(k)}),$$

which we condition on. Next we show that the lemma statement holds for $k = i$. By definition of $\boldsymbol{v}^{(i)}$ (Line 27 and 28),

$$\boldsymbol{v}^{(i-1)} \le \boldsymbol{v}^{(i)} \quad \text{and} \quad \boldsymbol{v}(\boldsymbol{Q}^{(i-1)}) \le \boldsymbol{v}^{(i)}.$$

Furthermore, since $\boldsymbol{v}^{(0)} \le \boldsymbol{v}^{(1)} \le \ldots \le \boldsymbol{v}^{(i-1)} \le \mathcal{T}_{\pi^{i-1}}\boldsymbol{v}^{(i-1)} \le \mathcal{T}\boldsymbol{v}^{(i-1)} \le \mathcal{T}^\infty\boldsymbol{v}^{(i-1)} = \boldsymbol{v}^*$, we have

$$\boldsymbol{v}^{(i)} - \boldsymbol{v}^{(0)} \le \boldsymbol{v}^* - \boldsymbol{v}^{(0)} \le u\mathbf{1}.$$

By Lemma 4.3, we have, with probability at least $1 - \delta'$

$$\boldsymbol{P}\big[\boldsymbol{v}^{(i)} - \boldsymbol{v}^{(0)}\big] - \frac{(1-\gamma)u}{8}\cdot\mathbf{1} \le \boldsymbol{g}^{(i)} \le \boldsymbol{P}\big[\boldsymbol{v}^{(i)} - \boldsymbol{v}^{(0)}\big], \tag{E.6}$$

which we condition on for the rest of the proof. Thus we have

$$\boldsymbol{Q}^{(i)} = \boldsymbol{r} + \gamma(\boldsymbol{w} + \boldsymbol{g}^{(i)}) \le \boldsymbol{r} + \gamma(\boldsymbol{P}\boldsymbol{v}^{(0)} + \boldsymbol{P}\boldsymbol{v}^{(i)} - \boldsymbol{P}\boldsymbol{v}^{(0)}) = \boldsymbol{r} + \gamma\boldsymbol{P}\boldsymbol{v}^{(i)}.$$

To show $\boldsymbol{v}^{(i)} \le \mathcal{T}_{\pi^{(i)}}\boldsymbol{v}^{(i)}$, we notice that if for some $s$, $\pi^{(i)}(s) \ne \pi^{(i-1)}(s)$, then

$$\boldsymbol{v}^{(i)}(s) \le [\mathcal{T}_{\pi^{(i)}}\boldsymbol{v}^{(i-1)}](s) \le [\mathcal{T}_{\pi^{(i)}}\boldsymbol{v}^{(i)}](s),$$

where the first inequality follows from $\boldsymbol{v}^{(i)}(s) \le \boldsymbol{r}(s, \pi^{(i)}(s)) + \gamma\boldsymbol{P}_{s,\pi^{(i)}(s)}^\top\boldsymbol{v}^{(i-1)} = \mathcal{T}_{\pi^{(i)}}\boldsymbol{v}^{(i-1)}$. On the other hand, if $\pi^{(i)}(s) = \pi^{(i-1)}(s)$, then

$$\boldsymbol{v}^{(i)}(s) = \boldsymbol{v}^{(i-1)}(s) \le (\mathcal{T}_{\pi^{(i-1)}}\boldsymbol{v}^{(i-1)})(s) \le (\mathcal{T}_{\pi^{(i-1)}}\boldsymbol{v}^{(i)})(s) = (\mathcal{T}_{\pi^{(i)}}\boldsymbol{v}^{(i)})(s).$$

This completes the induction step. Lastly, combining (E.5) and (E.6), we have

$$\begin{aligned}
\boldsymbol{Q}^* - \boldsymbol{Q}^{(i)} &= \boldsymbol{Q}^* - \boldsymbol{r} - \gamma(\boldsymbol{w} + \boldsymbol{g}^{(i)}) = \gamma\boldsymbol{P}\boldsymbol{v}(\boldsymbol{Q}^*) - \gamma(\boldsymbol{w} + \boldsymbol{g}^{(i)}) \\
&= \gamma\boldsymbol{P}\boldsymbol{v}(\boldsymbol{Q}^*) - \gamma\boldsymbol{P}(\boldsymbol{v}^{(i)} - \boldsymbol{v}^{(0)}) - \gamma\boldsymbol{P}\boldsymbol{v}^{(0)} + \boldsymbol{\xi}^{(i)} \\
&= \gamma\boldsymbol{P}\boldsymbol{v}(\boldsymbol{Q}^*) - \gamma\boldsymbol{P}\boldsymbol{v}^{(i)} + \boldsymbol{\xi}^{(i)},
\end{aligned}$$

where

$$\boldsymbol{\xi}^{(i)} \le (1-\gamma)u/8\cdot\mathbf{1} + 2\sqrt{2\alpha_1\boldsymbol{\sigma}_{\boldsymbol{v}^*}} + 2\sqrt{2\alpha_1}u\cdot\mathbf{1} + 16\alpha_1^{3/4}\|\boldsymbol{v}^{(0)}\|_\infty\cdot\mathbf{1} + (4/3)\cdot\alpha_1\cdot\|\boldsymbol{v}^{(0)}\|_\infty\cdot\mathbf{1},$$

where $\alpha_1 = \log(8|\mathcal{S}||\mathcal{A}|\delta^{-1})/m_1 \le 1$. Mover, since $\boldsymbol{v}(\boldsymbol{Q}^{(i-1)}) \le \boldsymbol{v}^{(i)}$, we obtain

$$\boldsymbol{Q}^* - \boldsymbol{Q}^{(i)} \le \gamma\boldsymbol{P}\boldsymbol{v}(\boldsymbol{Q}^*) - \gamma\boldsymbol{P}\boldsymbol{v}(\boldsymbol{Q}^{(i-1)}) + \boldsymbol{\xi}^{(i)} \le \gamma\boldsymbol{P}^{\pi^*}\boldsymbol{Q}^* - \gamma\boldsymbol{P}^{\pi^*}\boldsymbol{Q}^{(i-1)} + \boldsymbol{\xi}^{(i)},$$

where $\pi^*$ is an arbitrary optimal policy and we use the fact that $\max_a \boldsymbol{Q}^*(s, a) = \boldsymbol{Q}^*(s, \pi^*(s))$. This completes the proof of the lemma. $\qquad\square$

*Proof of Proposition 4.5.* Recall that we are able to sample a state from each $\boldsymbol{P}_{s,a}$ with time $O(1)$. Let $\beta = (1-\gamma)^{-1}$, $R = \lceil c_1\beta \ln[\beta u^{-1}]\rceil$, $m_1 = c_2\beta^3 u^{-2} \cdot \log(8|\mathcal{S}||\mathcal{A}|\delta^{-1})$ and $m_2 = c_3\beta^2 \cdot \log[2R|\mathcal{S}||\mathcal{A}|\delta^{-1}]$ for some constants $c_1, c_2$ and $c_3$ required in Algorithm 1. In the following proof, we set $c_1, c_2, c_3$ to be sufficiently large but otherwise arbitrary absolute constants (e.g., $c_1 \geq 4, c_2 \geq 8192, c_3 \geq 128$). By Lemma 4.4, with probability at least $1-2\delta$ for each $1 \leq i \leq R$, we have $\boldsymbol{v}^{(i-1)} \leq \boldsymbol{v}^{(i)} \leq \mathcal{T}_{\pi^{(i)}}\boldsymbol{v}^{(i)}$, and $\boldsymbol{Q}^{(i)} \leq \boldsymbol{r} + \gamma\boldsymbol{P}\boldsymbol{v}^{(i)}$,

$$\boldsymbol{Q}^* - \boldsymbol{Q}^{(i)} \leq \gamma\boldsymbol{P}^{\pi^*}\big[\boldsymbol{Q}^* - \boldsymbol{Q}^{(i-1)}\big] + \boldsymbol{\xi},$$

where

$$\boldsymbol{\xi} \leq (1-\gamma)u/C \cdot \mathbf{1} + C\sqrt{\alpha_1\boldsymbol{\sigma}_{\boldsymbol{v}^*}} + C\alpha_1^{3/4}\|\boldsymbol{v}^{(0)}\|_\infty \cdot \mathbf{1}$$

for $\alpha_1 = \log(8|\mathcal{S}||\mathcal{A}|\delta^{-1})/m_1$ and sufficiently large constant $C$. Solving the recursion, we obtain

$$\boldsymbol{Q}^* - \boldsymbol{Q}^{(R-1)} \leq \gamma^{R-1}\boldsymbol{P}^{\pi^*}\big[\boldsymbol{Q}^* - \boldsymbol{Q}^0\big] + \sum_{i=0}^{R-1}\gamma^i(\boldsymbol{P}^{\pi^*})^i\boldsymbol{\xi}$$

$$\leq \gamma^{R-1}\boldsymbol{P}^{\pi^*}\big[\boldsymbol{Q}^* - \boldsymbol{Q}^0\big] + (\boldsymbol{I} - \gamma\boldsymbol{P}^{\pi^*})^{-1}\boldsymbol{\xi}.$$

We first apply a naïve bound $\|\boldsymbol{P}^{\pi^*}[\boldsymbol{Q}^* - \boldsymbol{Q}^0]\|_\infty \leq (1-\gamma)^{-1}$. Hence

$$\gamma^{R-1}\boldsymbol{P}^{\pi^*}\big[\boldsymbol{Q}^* - \boldsymbol{Q}^0\big] \leq \frac{u}{4} \cdot \mathbf{1},$$

where $R = \lceil(1-\gamma)^{-1}\ln[4(1-\gamma)^{-1}u^{-1}]\rceil + 1$. The next step is the key to the improvement in our analysis. We further apply the bound in Lemma C.1, given by

$$(\boldsymbol{I} - \gamma\boldsymbol{P}^{\pi^*})^{-1}\sqrt{\boldsymbol{\sigma}_{\boldsymbol{v}^*}} \leq \min(2\gamma^{-1}(1-\gamma)^{-1.5}, (1-\gamma)^{-2}) \cdot \mathbf{1} \leq 3(1-\gamma)^{-1.5} \cdot \mathbf{1},$$

where the last inequality follows since $\min(2\gamma^{-1}, (1-\gamma)^{-1/2}) \leq 3$. With $\|(\boldsymbol{I} - \gamma\boldsymbol{P}^{\pi^*})^{-1}\mathbf{1}\|_\infty \leq (1-\gamma)^{-1}$ and $\|\boldsymbol{v}^{(0)}\|_\infty \leq (1-\gamma)^{-1}$, we have,

$$(\boldsymbol{I} - \gamma\boldsymbol{P}^{\pi^*})^{-1}\boldsymbol{\xi} \leq \left[\frac{u}{8} + C'\sqrt{\frac{2\alpha_1}{\gamma^2(1-\gamma)^3}} + C'\frac{\alpha_1^{3/4}}{(1-\gamma)^2}\right] \cdot \mathbf{1}$$

$$\leq \left[\frac{u}{8} + \frac{u}{16} + \left(\frac{(1-\gamma)^3 u^2}{C'' \cdot (1-\gamma)^{8/3}}\right)^{3/4}\right] \cdot \mathbf{1}$$

$$\leq \frac{u}{4} \cdot \mathbf{1},$$

for some sufficiently large $C'$ and $C''$, which depend on $c_1, c_2$ and $c_3$. Since $\boldsymbol{v}(\boldsymbol{Q}^{(R-1)}) \leq \boldsymbol{v}^{(R)}$, we have

$$\boldsymbol{v}^* - \boldsymbol{v}^{(R)} \leq \boldsymbol{v}^* - \boldsymbol{v}(\boldsymbol{Q}^{(R-1)}) \leq \gamma^{R-1}\boldsymbol{P}^{\pi^*}\big[\boldsymbol{Q}^* - \boldsymbol{Q}^0\big] + (\boldsymbol{I} - \gamma\boldsymbol{P}^{\pi^*})^{-1}\boldsymbol{\xi} \leq \frac{u}{2} \cdot \mathbf{1}.$$

This completes the proof of the correctness. It remains to bound the time complexity. The initialization stage costs $O(m_1)$ time per $(s, a)$. Each iteration costs $O(m_2)$ time per $(s, a)$. We thus have the total time complexity as

$$O(m_1 + Rm_2)|\mathcal{S}|||\mathcal{A}|| = O\left[\frac{|\mathcal{S}||\mathcal{A}|}{(1-\gamma)^3} \cdot \log\frac{|\mathcal{S}||\mathcal{A}|}{\delta \cdot (1-\gamma) \cdot u} \cdot \left(\frac{1}{u^2} + \log\frac{1}{(1-\gamma) \cdot u}\right)\right].$$

Since $\log[(1-\gamma)^{-1}u^{-1}] = O(\log[(1-\gamma)^{-1}]u^{-2})$, we conclude the proof. $\qquad\square$

### E.2 Missing Analysis of Halving Errors

We refer in this section to Algorithm 1 as a subroutine HALFERR, which given an input MDP $\mathcal{M}$ with a sampling oracle, an input value function $\boldsymbol{v}^{(i)}$ and an input policy $\pi^{(i)}$, outputs an value function $\boldsymbol{v}^{(i+1)}$ and a policy $\pi^{(i+1)}$ such that, with high probability (over the new samples of the sampling oracle),

$$\|\boldsymbol{Q}^{(i+1)} - \boldsymbol{Q}^*\|_\infty \leq \|\boldsymbol{Q}^{(i)} - \boldsymbol{Q}^*\|_\infty/2 \quad \text{and} \quad \|\boldsymbol{v}^{\pi^{(i+1)}} - \boldsymbol{v}^*\|_\infty \leq \|\boldsymbol{v}^{\pi^{(i)}} - \boldsymbol{v}^*\|_\infty/2.$$

After $\log[\epsilon^{-1}(1-\gamma)^{-1}]$ calls of the subroutine HALFERR, the final output policy and value functions are $\epsilon$-close to the optimal ones with high probability.

We summarize our meta algorithm in Algorithm 2. Note that in the algorithm, each call of HALFERR will draw new samples from the sampling oracle. These new samples guarantee the independence of successive improvements and also save space of the algorithm. For instance, the algorithm HAL-FERR only needs to use $O(|\mathcal{S}||\mathcal{A}|)$ words of memory instead of storing all the samples. The guarantee of the algorithm is summarized in Proposition E.3.

**Proposition E.3.** Let $\mathcal{M} = (\mathcal{S}, \mathcal{A}, \boldsymbol{r}, \boldsymbol{P}, \gamma)$ with a sampling oracle. Suppose HALFERR is an algorithm that takes an input $\boldsymbol{v}^{(i)}$ and an input policy $\pi^{(i)}$ and a number $u \in [0, (1-\gamma)^{-1}]$ satisfying $\boldsymbol{v}^* - u\boldsymbol{1} \leq \boldsymbol{v}^{(i)} \leq \boldsymbol{v}^{\pi^{(i)}}$, halts in time $\tau$ and outputs a $\boldsymbol{v}^{(i+1)}$ and a policy $\pi^{(i+1)}$ satisfying,

$$\boldsymbol{v}^* - \frac{u}{2} \cdot \boldsymbol{1} \leq \boldsymbol{v}^{(i+1)} \leq \boldsymbol{v}^{\pi^{(i+1)}} \leq \boldsymbol{v}^*.$$

with probability at least $1 - (1 - \gamma) \cdot \epsilon \cdot \delta$ (over the randomness of the new samples given by the sampling oracle), then the meta algorithm described in Algorithm 2, given input $\mathcal{M}$ and the sampling oracle, halts in $\tau \cdot \log(\epsilon^{-1} \cdot (1-\gamma)^{-1})$ and outputs an policy $\pi^{(R)}$ such that

$$\boldsymbol{v}^* - \epsilon \cdot \boldsymbol{1} \leq \boldsymbol{v}^{(R)} \leq \boldsymbol{v}^{\pi^{(R)}} \leq \boldsymbol{v}^*$$

with probability at least $1 - \delta$ (over the randomness of all samples drawn from the sampling oracle). Moreover, if HALFERR uses space $s$, then the meta algorithm uses space $s + O(|\mathcal{S}||\mathcal{A}|)$. If each call of HALFERR takes $m$ samples from the oracle, then the overall samples taken by Algorithm 2 is $m \cdot \log(\epsilon^{-1} \cdot (1-\gamma)^{-1})$.

The proof of this proposition is a straightforward application of conditional probability.

*Proof of Proposition E.3.* The proof follows from a straightforward induction. For simplicity, denote $\beta = (1-\gamma)^{-1}$. In the meta-algorithm, the initialization is $\boldsymbol{v}^{(0)} = \boldsymbol{0}$ and $\pi^{(0)}$ is an arbitrary policy. Thus $\boldsymbol{v}^* - \beta \cdot \boldsymbol{1} \leq \boldsymbol{v}^{(0)} \leq \boldsymbol{v}^{\pi^{(0)}}$. By running the meta-algorithm, we obtain a sequence of value functions and policies: $\{\boldsymbol{v}^{(i)}\}_{i=0}^R$ and $\{\pi^{(i)}\}_{i=0}^R$. Since each call of the HALFERR uses new samples from the oracle, the sequence of value functions and policies satisfies strong Markov property (given $(\boldsymbol{v}^{(i)}, \pi^{(i)})$, $(\boldsymbol{v}^{(i+1)}, \pi^{(i+1)})$ is independent with $\{(\boldsymbol{v}^{(j)}, \pi^{(j)})\}_{j=0}^{i-1}$). Thus

$$\Pr\left[\boldsymbol{v}^* - 2^{-R}\beta \cdot \boldsymbol{1} \leq \boldsymbol{v}^{(R)} \leq \boldsymbol{v}^{\pi^{(R)}}\right]$$
$$\geq \prod_{i=1}^R \Pr\left[\boldsymbol{v}^* - 2^{-i}\beta \cdot \boldsymbol{1} \leq \boldsymbol{v}^{(i)} \leq \boldsymbol{v}^{\pi^{(i)}} \Big| \boldsymbol{v}^* - 2^{-i+1}\beta \cdot \boldsymbol{1} \leq \boldsymbol{v}^{(i-1)} \leq \boldsymbol{v}^{\pi^{(i-1)}}\right]$$
$$\geq 1 - \delta.$$

Since $2^{-R}(1-\gamma)^{-1} \leq \epsilon$, we conclude the proof. $\qquad\square$

*Proof of Theorem 4.6.* Our algorithm is simply plugging in Algorithm 1 as the HALFERR subroutine in Algorithm 2. The correctness is guaranteed by Proposition E.3 and Proposition 4.5. The running time guarantee follows from a straightforward calculation. $\qquad\square$

# F   Extension to Finite Horizon

In this section we show how to apply similar techniques to achieve improved sample complexities for solving finite Horizon MDPs given a generative model and we prove that the sample complexity we achieve is optimal up to logarithmic factors.

The finite horizon problem is to compute an optimal non-stationary policy over a fixed time horizon $H$, i.e. a policy of the form $\pi(s, h)$ for $s \in S$ and $h \in \{0, \ldots H\}$), where the reward is the expected cumulative (un-discounted) reward for following this policy. In classic value iteration, this is typically done using a backward recursion from time $H, H - 1, \ldots 0$. We show how to use the ideas in this paper to solve for an $\epsilon$-approximate policy. As we have shown in the discounted case, it is suffice to show an algorithm that decrease the error of the value at each stage by half. Our algorihtm is presented in Algorithm 3.

To analyze the algorithm, we first provide an analogous lemma of Lemma 4.1,

**Lemma F.1** (Empirical Estimation Error). *Let $\widetilde{\boldsymbol{w}}_h$ and $\widehat{\boldsymbol{\sigma}}_h$ be computed in Line 10 of Algorithm 3. Recall that $\widetilde{\boldsymbol{w}}_h$ and $\widehat{\boldsymbol{\sigma}}_h$ are empirical estimates of $\boldsymbol{Pv}_h$ and $\boldsymbol{\sigma}_{\boldsymbol{v}_h} = \boldsymbol{Pv}_h^2 - (\boldsymbol{Pv}_h)^2$ using $m_1$ samples per $(s,a)$ pair. Then with probability at least $1 - \delta$, for $L \stackrel{\text{def}}{=} \log(8|\mathcal{S}||\mathcal{A}|\delta^{-1})$ and every $h = 1, 2, \ldots, H$, we have*

$$\left|\widetilde{\boldsymbol{w}}_h - \boldsymbol{P}^\top \boldsymbol{v}_h^{(0)}\right| \leq \sqrt{2m_1^{-1}\boldsymbol{\sigma}_{\boldsymbol{v}_h^{(0)}} \cdot L} + 2(3m_1)^{-1}\|\boldsymbol{v}_h^{(0)}\|_\infty L \tag{F.1}$$

*and*

$$\forall (s,a) \in \mathcal{S} \times \mathcal{A}: \quad \left|\widehat{\boldsymbol{\sigma}}_h(s,a) - \boldsymbol{\sigma}_{\boldsymbol{v}_h^{(0)}}(s,a)\right| \leq 4\|\boldsymbol{v}_h^{(0)}\|_\infty^2 \cdot \sqrt{2m_1^{-1}L}. \tag{F.2}$$

*Proof.* The proof of this lemma is identical to that of Lemma 4.1. $\square$

An analogous lemma to Lemma 4.3 is also presented here.

**Lemma F.2.** *Let $\boldsymbol{g}_h^{(i)}$ be the estimate of $\boldsymbol{P}\big[\boldsymbol{v}_h^{(i)} - \boldsymbol{v}_h^{(0)}\big]$ defined in Line 27 of Algorithm 3. Then conditioning on the event that $\|\boldsymbol{v}_h^{(i)} - \boldsymbol{v}_h^{(0)}\|_\infty \leq 2\epsilon$, with probability at least $1 - \delta/H$,*

$$\boldsymbol{P}\big[\boldsymbol{v}_h^{(i)} - \boldsymbol{v}_h^{(0)}\big] - \frac{\epsilon}{4H} \cdot \mathbf{1} \leq \boldsymbol{g}_h^{(i)} \leq \boldsymbol{P}\big[\boldsymbol{v}_h^{(i)} - \boldsymbol{v}_h^{(0)}\big]$$

*provided appropriately chosen constants in Algorithm 3.*

*Proof.* The proof of this lemma is identical to that of Lemma 4.3 except that $(1 - \gamma)^{-1}$ is replaced with $H$. $\square$

Similarly, we can show the following improvement lemma.

**Lemma F.3.** *Let $\boldsymbol{Q}_h$ be the estimated Q-function of $\boldsymbol{v}_{h+1}$ in Line 30 of Algorithm 3. Let $\boldsymbol{Q}_h^* = \boldsymbol{r} + \boldsymbol{P}_h \boldsymbol{v}_{h+1}^*$ be the optimal Q-function of the DMDP. Let $\pi(\cdot, h)$ and $\boldsymbol{v}_h$ be estimated in iteration $h$, as defined in Line 24 and 25. Let $\pi^*$ be an optimal policy for the DMDP. For a policy $\pi$, let $\boldsymbol{P}_h^\pi \boldsymbol{Q} \in \mathbb{R}^{\mathcal{S} \times \mathcal{A}}$ be defined as $(\boldsymbol{P}_h^\pi \boldsymbol{Q})(s,a) = \sum_{s' \in \mathcal{S}} \boldsymbol{P}_{s,a}(s')\boldsymbol{Q}(s', \pi(s', h))$. Suppose for all $h \in [H-1]$, $\boldsymbol{v}_h^{(0)} \leq \mathcal{T}_{\pi^{(0)}(\cdot, h)}\boldsymbol{v}_{h+1}^{(0)}$. Let $\boldsymbol{v}_{H+1} \stackrel{\text{def}}{=} \mathbf{0}$ and $\boldsymbol{Q}_{H+1} \stackrel{\text{def}}{=} 0$. Then, with probability at least $1 - 2\delta$, for all $1 \leq h \leq H$, $\boldsymbol{v}_h^{(0)} \leq \boldsymbol{v}_h \leq \mathcal{T}_{\pi(\cdot,h)}\boldsymbol{v}_{h+1} \leq \boldsymbol{v}_h^*$, $\boldsymbol{Q}_h \leq \boldsymbol{r} + \boldsymbol{P}_h \boldsymbol{v}_{h+1}$ and*

$$\boldsymbol{Q}_h^* - \boldsymbol{Q}_h \leq \boldsymbol{P}_h^{\pi^*}\big[\boldsymbol{Q}_{h+1}^* - \boldsymbol{Q}_{h+1}\big] + \boldsymbol{\xi}_h,$$

*where the error vector $\boldsymbol{\xi}_h$ satisfies*

$$\mathbf{0} \leq \boldsymbol{\xi}_h \leq 8H^{-1}u \cdot \mathbf{1} + 2\sqrt{2\alpha_1 \boldsymbol{\sigma}_{\boldsymbol{v}_{h+1}^*}} + 2\sqrt{2\alpha_1}u \cdot \mathbf{1} + 16\alpha_1^{3/4}\|\boldsymbol{v}_{h+1}^{(0)}\|_\infty \cdot \mathbf{1} + (4/3) \cdot \alpha_1 \cdot \|\boldsymbol{v}_{h+1}^{(0)}\|_\infty \cdot \mathbf{1},$$

*and $\alpha_1 = \log(8|\mathcal{S}||\mathcal{A}|H\delta^{-1})/m_1$.*

*Proof of Lemma F.3.* By Lemma 4.1, for any $h = 1, 2, \ldots, H$, with probability at least $1 - \delta/H$,

$$\left|\widetilde{\boldsymbol{w}}_h - \boldsymbol{P}\boldsymbol{v}_{h+1}\right| \leq \sqrt{2\alpha_1 \boldsymbol{\sigma}_{\boldsymbol{v}_{h+1}^{(0)}}} + \frac{2}{3} \cdot \alpha_1 \cdot \|\boldsymbol{v}_{h+1}^{(0)}\|_\infty \cdot \mathbf{1},$$

and

$$\left|\widehat{\boldsymbol{\sigma}}_{h+1} - \boldsymbol{\sigma}_{\boldsymbol{v}_{h+1}^{(0)}}\right| \leq 4\|\boldsymbol{v}_{h+1}^{(0)}\|_\infty^2 \cdot \sqrt{2\alpha_1} \cdot \mathbf{1}, \tag{F.3}$$

which we condition on. We have

$$\left|\widetilde{\boldsymbol{w}}_h - \boldsymbol{P}\boldsymbol{v}_{h+1}^{(0)}\right| \leq \sqrt{2\alpha_1 \widehat{\boldsymbol{\sigma}}_{h+1}} + (4\alpha_1^{3/4}\|\boldsymbol{v}_{h+1}^{(0)}\|_\infty + \frac{2}{3} \cdot \alpha_1 \cdot \|\boldsymbol{v}_{h+1}^{(0)}\|_\infty)\mathbf{1}.$$

Thus

$$\boldsymbol{w}_h = \widetilde{\boldsymbol{w}}_h - \sqrt{2\alpha_1 \widehat{\boldsymbol{\sigma}}_{h+1}} - 4\alpha_1^{3/4}\|\boldsymbol{v}_{h+1}^{(0)}\|_\infty \mathbf{1} - \frac{2}{3} \cdot \alpha_1 \cdot \|\boldsymbol{v}_{h+1}^{(0)}\|_\infty \mathbf{1} \leq \boldsymbol{P}\boldsymbol{v}_{h+1}^{(0)}, \tag{F.4}$$

and

$$\boldsymbol{w}_h \geq \boldsymbol{P}\boldsymbol{v}_{h+1}^{(0)} - 2\sqrt{2\alpha_1 \widehat{\boldsymbol{\sigma}}_{h+1}} - (8\alpha_1^{3/4}\|\boldsymbol{v}_{h+1}^{(0)}\|_\infty + \frac{4}{3} \cdot \alpha_1 \cdot \|\boldsymbol{v}_{h+1}^{(0)}\|_\infty)\mathbf{1}.$$

By (E.3) and Lemma 4.2, we have

$$\sqrt{\widehat{\boldsymbol{\sigma}}_{h+1}} \leq \sqrt{\boldsymbol{\sigma}_{\boldsymbol{v}_{h+1}^{(0)}}} + 2\|\boldsymbol{v}_{h+1}^{(0)}\|_{\infty}(2\alpha)^{1/4}\mathbf{1} \leq \sqrt{\boldsymbol{\sigma}_{\boldsymbol{v}_{h+1}^{*}}} + \epsilon\mathbf{1} + 2\|\boldsymbol{v}_{h+1}^{(0)}\|_{\infty}(2\alpha)^{1/4}\mathbf{1}.$$

we have

$$\boldsymbol{w}_h \geq \boldsymbol{P}\boldsymbol{v}_{h+1}^{(0)} - 2\sqrt{2\alpha_1\boldsymbol{\sigma}_{\boldsymbol{v}_{h+1}^{*}}} - 2\sqrt{2\alpha_1}\epsilon\mathbf{1} - 16\alpha_1^{3/4}\|\boldsymbol{v}_{h+1}^{(0)}\|_{\infty}\mathbf{1} - \frac{4}{3}\cdot\alpha_1\cdot\|\boldsymbol{v}_{h+1}^{(0)}\|_{\infty}\mathbf{1} \quad \text{(F.5)}$$

For the rest of the proof, we condition on the event that (F.4) and (F.5) hold for all $h = 1, 2, \ldots, H$, which happens with probability at least $1 - \delta$. Denote $\boldsymbol{v}_{H+1}^{*} = \boldsymbol{v}_{H+1} = \boldsymbol{v}_{H+1}^{(0)} = \mathbf{0}$. Thus we have $\boldsymbol{v}_{H+1}^{(0)} \leq \boldsymbol{v}_{H+1} \leq \boldsymbol{v}_{H+1}^{*}$. Next we prove the lemma by induction on $h$. Assume for some $h$, with probability at least $1 - (h-1)\delta/H$ the following holds, for all $h' = h+1, h+2, \ldots, H$,

$$\boldsymbol{v}_{h'}^{(0)} \leq \boldsymbol{v}_{h'} \leq \boldsymbol{v}_{h'}^{*},$$

which we condition on. Next we show that the lemma statement holds for $h$ as well. By definition of $\boldsymbol{v}_h$ (Line 27 and 28),

$$\boldsymbol{v}_h^{(0)} \leq \boldsymbol{v}_h \quad \text{and} \quad \boldsymbol{v}(\boldsymbol{Q}_h) \leq \boldsymbol{v}_h.$$

Furthermore, since $\boldsymbol{v}_{h+1}^{(0)} \leq \boldsymbol{v}_{h+1}^{*} \leq \boldsymbol{v}_{h+1}^{(0)} + u\mathbf{1}$ we have

$$\boldsymbol{v}_{h+1}^{*} - \boldsymbol{v}_{h+1} \leq \boldsymbol{v}_{h+1}^{*} - \boldsymbol{v}_{h+1}^{(0)} \leq u\mathbf{1}.$$

By Lemma 4.3, we have, with probability at least $1 - \delta'$

$$\boldsymbol{P}\big[\boldsymbol{v}_{h+1} - \boldsymbol{v}_{h+1}^{(0)}\big] - \frac{u}{8H}\cdot\mathbf{1} \leq \boldsymbol{g}_h \leq \boldsymbol{P}\big[\boldsymbol{v}_{h+1} - \boldsymbol{v}_{h+1}^{(0)}\big], \quad \text{(F.6)}$$

which we condition on for the rest of the proof. Thus we have

$$\boldsymbol{Q}_h = \boldsymbol{r} + (\boldsymbol{w}_h + \boldsymbol{g}_h) \leq \boldsymbol{r} + \boldsymbol{P}\boldsymbol{v}_{h+1}^{(0)} + \boldsymbol{P}\boldsymbol{v}_{h+1} - \boldsymbol{P}\boldsymbol{v}_{h+1}^{(0)} = \boldsymbol{r} + \boldsymbol{P}\boldsymbol{v}_{h+1} \leq \boldsymbol{Q}_h^{*}.$$

To show $\boldsymbol{v}_h \leq \mathcal{T}_{\pi(\cdot,h)}\boldsymbol{v}_{h+1}$, we notice that if for some $s$, $\pi(s,h) \neq \pi^{(0)}(s,h)$, then,

$$\boldsymbol{v}_h(s) \leq \boldsymbol{r}(s,\pi(s,h)) + \boldsymbol{P}_{s,\pi(s,h)}^{\top}\boldsymbol{v}_{h+1} = \mathcal{T}_{\pi(\cdot,h)}\boldsymbol{v}_{h+1}.$$

On the other hand, if $\pi(s,h) = \pi^{(0)}(s,h)$, then

$$\forall s \in \mathcal{S}: \quad \boldsymbol{v}_h(s) = \boldsymbol{v}_h^{(0)}(s) \leq (\mathcal{T}_{\pi^{(0)}(\cdot,h)}\boldsymbol{v}_{h+1}^{(0)})(s) \leq (\mathcal{T}_{\pi^{(0)}(\cdot,h)}\boldsymbol{v}_{h+1})(s) = (\mathcal{T}_{\pi(\cdot,h)}\boldsymbol{v}_{h+1})(s).$$

This completes the induction step. Lastly, combining (F.5) and (F.6), we have

$$\begin{aligned}
\boldsymbol{Q}_h^{*} - \boldsymbol{Q}_h &= \boldsymbol{Q}_h^{*} - \boldsymbol{r} - (\boldsymbol{w}_h + \boldsymbol{g}_h) = \boldsymbol{P}\boldsymbol{v}(\boldsymbol{Q}_{h+1}^{*}) - (\boldsymbol{w}_h + \boldsymbol{g}_h) \\
&= \boldsymbol{P}\boldsymbol{v}(\boldsymbol{Q}_{h+1}^{*}) - \boldsymbol{P}(\boldsymbol{v}_{h+1} - \boldsymbol{v}_{h+1}^{(0)}) - \boldsymbol{P}\boldsymbol{v}_{h+1} + \boldsymbol{\xi}_h \\
&= \boldsymbol{P}\boldsymbol{v}(\boldsymbol{Q}_{h+1}^{*}) - \boldsymbol{P}\boldsymbol{v}_{h+1} + \boldsymbol{\xi}_h,
\end{aligned}$$

where

$$\boldsymbol{\xi}_h \leq H^{-1}u/8\cdot\mathbf{1} + 2\sqrt{2\alpha_1\boldsymbol{\sigma}_{\boldsymbol{v}_{h+1}^{*}}} + 2\sqrt{2\alpha_1}u\cdot\mathbf{1} + 16\alpha_1^{3/4}\|\boldsymbol{v}_{h+1}^{(0)}\|_{\infty}\cdot\mathbf{1} + (4/3)\cdot\alpha_1\cdot\|\boldsymbol{v}_{h+1}^{(0)}\|_{\infty}\cdot\mathbf{1},$$

where $\alpha_1 = \log(8|\mathcal{S}||\mathcal{A}|\delta^{-1})/m_1$. Mover, since $\boldsymbol{v}(\boldsymbol{Q}_{h+1}) \leq \boldsymbol{v}_{h+1}$, we obtain

$$\boldsymbol{Q}_h^{*} - \boldsymbol{Q}_h \leq \boldsymbol{P}\boldsymbol{v}(\boldsymbol{Q}_{h+1}^{*}) - \boldsymbol{P}\boldsymbol{v}(\boldsymbol{Q}_{h+1}) + \boldsymbol{\xi}_h \leq \boldsymbol{P}_h^{\pi^{*}}\boldsymbol{Q}_{h+1}^{*} - \boldsymbol{P}_h^{\pi^{*}}\boldsymbol{Q}_{h+1} + \boldsymbol{\xi}_h,$$

where $\pi^{*}$ is an arbitrary optimal policy and we use the fact that $\max_a \boldsymbol{Q}_h^{*}(s,a) = \boldsymbol{Q}_h^{*}(s,\pi^{*}(s,h))$. This completes the proof of the lemma. $\qquad\square$

Furthermore, we show an analogous lemma of Lemma C.1.

**Lemma F.4** (Upper Bound on Variance). *For any $\pi$, we have*

$$\bigg\|\sum_{h'=h}^{H-1}\bigg(\prod_{i=h+1}^{h'}\boldsymbol{P}_i^{\pi}\bigg)\sqrt{\boldsymbol{\sigma}_{\boldsymbol{v}_{h'+1}^{\pi}}}\bigg\|_{\infty}^2 \leq H^{3/2}.$$

*Proof.* First, by Cauchy-Swartz inequality, we have

$$\sum_{h'=h}^{H-1}\left(\prod_{i=h+1}^{h'}\boldsymbol{P}_i^{\pi}\right)\sqrt{\boldsymbol{\sigma}_{\boldsymbol{v}_{h'+1}^{\pi}}} \leq \sqrt{H\sum_{h'=h}^{H-1}\left(\prod_{i=h+1}^{h'}\boldsymbol{P}_i^{\pi}\right)\boldsymbol{\sigma}_{\boldsymbol{v}_{h'+1}^{\pi}}}.$$

Next, by a similar argument of the proof of Lemma C.2, we can show that

$$\left[\sum_{h'=h}^{H-1}\left(\prod_{i=h+1}^{h'}\boldsymbol{P}_i^{\pi}\right)\boldsymbol{\sigma}_{\boldsymbol{v}_{h'+1}^{\pi}}\right](s) = \mathrm{Var}\left[\sum_{t=h}^{H}r(s^t,\pi(s^t,t))\Big|s^h=s\right] \leq H^2.$$

This completes the proof. $\qquad\square$

We are now ready to present the guarantee of the algorithm 3.

**Proposition F.5.** *On an input value vectors $\boldsymbol{v}_1^{(0)},\boldsymbol{v}_2^{(0)},\ldots,\boldsymbol{v}_H^{(0)}$, policy $\pi^{(0)}$, and parameters $u\in(0,\beta], \delta\in(0,1)$ such that $\boldsymbol{v}_h^{(0)}\leq\mathcal{T}_{\pi^{(0)}(\cdot,h)}\boldsymbol{v}_{h+1}^{(0)}$ for all $h\in[H-1]$, and $\boldsymbol{v}_h^{(0)}\leq\boldsymbol{v}_h^*\leq\boldsymbol{v}_h^{(0)}+u\mathbf{1}$, Algorithm 3 halts in time $O[u^{-2}\cdot H^4|\mathcal{S}||\mathcal{A}|\cdot\log(|\mathcal{S}||\mathcal{A}\delta^{-1}Hu^{-1})]$ and outputs $\boldsymbol{v}_1,\boldsymbol{v}_2,\ldots,\boldsymbol{v}_H$ and $\pi:\mathcal{S}\times[H]\to\mathcal{A}$ such that*

$$\forall h\in[H]: \quad \boldsymbol{v}_h\leq\mathcal{T}_{\pi(\cdot,h)}(\boldsymbol{v}_{h+1}) \quad\text{and}\quad \mathbf{0}\leq\boldsymbol{v}_h^*-\boldsymbol{v}_h\leq(u/2)\cdot\mathbf{1}$$

*with probability at least $1-\delta$, provided appropriately chosen constants, $c_1,c_2$ and $c_3$, in Algorithm 3. Moreover, the algorithm uses $O[u^{-2}\cdot H^3|\mathcal{S}||\mathcal{A}|\cdot\log(|\mathcal{S}||\mathcal{A}\delta^{-1}Hu^{-1})]$ samples from the sampling oracle.*

*Proof of Proposition F.5.* Recall that we are able to sample a state from each $\boldsymbol{P}_{s,a}$ with time $O(1)$. Let $R=\lceil c_1H\ln[Hu^{-1}]\rceil, m_1=c_2H^3u^{-2}\cdot\log(8|\mathcal{S}||\mathcal{A}|\delta^{-1})$ and $m_2=c_3H^2\cdot\log[2R|\mathcal{S}||\mathcal{A}|\delta^{-1}]$ for some constants $c_1,c_2$ and $c_3$ required in Algorithm 1. In the following proof, we set $c_1=4,c_2=8192,c_3=128$. By Lemma 4.4, with probability at least $1-2\delta$ for each $1\leq h\leq H$, we have $\boldsymbol{v}_h^{(0)}\leq\boldsymbol{v}_h\leq\mathcal{T}_{\pi(\cdot,h)}\boldsymbol{v}_h$, and $\boldsymbol{Q}_h\leq\boldsymbol{r}+\boldsymbol{P}\boldsymbol{v}_{h+1}$,

$$\boldsymbol{Q}_h^*-\boldsymbol{Q}_h\leq\boldsymbol{P}_h^{\pi^*}\left[\boldsymbol{Q}_{h+1}^*-\boldsymbol{Q}_{h+1}\right]+\boldsymbol{\xi}_h,$$

where

$$\boldsymbol{\xi}_h\leq H^{-1}u/8\cdot\mathbf{1}+2\sqrt{2\alpha_1\boldsymbol{\sigma}_{\boldsymbol{v}_{h+1}^*}}+2\sqrt{2\alpha_1}u\cdot\mathbf{1}+16\alpha_1^{3/4}\|\boldsymbol{v}_{h+1}^{(0)}\|_\infty\cdot\mathbf{1}+(4/3)\cdot\alpha_1\cdot\|\boldsymbol{v}_{h+1}^{(0)}\|_\infty\cdot\mathbf{1},$$

and $\alpha_1=\log(8|\mathcal{S}||\mathcal{A}|\delta^{-1})/m_1$. Notice that $\boldsymbol{v}_H^{(0)}=\boldsymbol{v}_H^*=\boldsymbol{v}(\boldsymbol{r})$, thus the $\boldsymbol{v}_H-\boldsymbol{v}_H^*=\mathbf{0}$. Solving the recursion, we obtain

$$\boldsymbol{Q}_h^*-\boldsymbol{Q}_h\leq\sum_{h'=h}^{H-1}\left(\prod_{i=h+1}^{h'}\boldsymbol{P}_i^{\pi^*}\right)\boldsymbol{\xi}_{h'}.$$

The next step is the key to the improvement in our analysis. We further apply the bound in Lemma C.1, given by

$$\sum_{h'=h}^{H-1}\left(\prod_{i=h+1}^{h'}\boldsymbol{P}_i^{\pi^*}\right)\sqrt{\boldsymbol{\sigma}_{\boldsymbol{v}_{h'+1}^*}}\leq H^{3/2}\cdot\mathbf{1}.$$

With $\|\sum_{h'=h}^{H-1}\prod_{i=h+1}^{h'}\boldsymbol{P}_i^{\pi^*}\mathbf{1}\|_\infty\leq H-h+1$ and $\|\boldsymbol{v}_h^{(0)}\|_\infty\leq H$, we have,

$$\sum_{h'=h}^{H-1}\left(\prod_{i=h+1}^{h'}\boldsymbol{P}_i^{\pi^*}\right)\boldsymbol{\xi}_{h'}\leq\left[\frac{u}{8}+4\sqrt{2\alpha_1H^3}+2H\sqrt{2\alpha_1}u+16H^2\alpha_1^{3/4}+\frac{4\alpha_1H^2}{3}\right]\mathbf{1}$$

$$\leq\left[\frac{u}{8}+\frac{u}{16}+\frac{\sqrt{H^{-1}}u}{32}+16\left(\frac{H^{-3}u^2}{32\cdot256\cdot(H)^{-8/3}}\right)^{3/4}+\frac{4H^{-1}u^2}{24\cdot256}\right]\cdot\mathbf{1}$$

$$\leq\frac{u}{4}\cdot\mathbf{1},$$

provided

$$\alpha_1 = \frac{\log(8|\mathcal{S}||\mathcal{A}|\delta^{-1})}{m_1} = c_2^{-1}H^3 u^{-2} \le \frac{H^{-3}u^2}{32 \cdot 256}.$$

Since $\boldsymbol{v}(\boldsymbol{Q}_h) \le \boldsymbol{v}_h$, we have

$$\boldsymbol{v}_h^* - \boldsymbol{v}_h \le \boldsymbol{v}^* - \boldsymbol{v}(\boldsymbol{Q}_h) \le \sum_{h'=h}^{H-1} \left( \prod_{i=h+1}^{h'} \boldsymbol{P}_i^{\pi^*} \right) \boldsymbol{\xi}_{h'} \le \frac{u}{2} \cdot \mathbf{1}.$$

This completes the proof of the correctness. It remains to bound the time complexity. The initialization stage costs $O(m_1)$ time per $(s,a)$ per stage $h$. Each iteration costs $O(m_2)$ time per $(s,a)$. We thus have the total time complexity as

$$O(Hm_1 + Hm_2)|\mathcal{S}||\mathcal{A}| = O\left[ H^4 \cdot |\mathcal{S}||\mathcal{A}| \cdot \log \frac{H|\mathcal{S}||\mathcal{A}|}{\delta \cdot u} \cdot \frac{1}{u^2} \right].$$

The total number of samples used is

$$O(m_1 + Hm_2)|\mathcal{S}||\mathcal{A}| = O\left[ H^3 \cdot |\mathcal{S}||\mathcal{A}| \cdot \log \frac{H|\mathcal{S}||\mathcal{A}|}{\delta \cdot u} \cdot \frac{1}{u^2} \right].$$

This completes the proof. $\qquad\square$

We can then use our meta-algorithm and obtain the following theorem.

**Theorem F.6.** *Let $\mathcal{M} = (\mathcal{S}, \mathcal{A}, \boldsymbol{P}, \boldsymbol{r}, H)$ be a $H$-MDP with a sampling oracle. Suppose we can sample a state from each probability vector $\boldsymbol{P}_{s,a}$ within time $O(1)$. Then there exists an algorithm that runs in time*

$$O\left[ \frac{1}{\epsilon^2} \cdot H^4 |\mathcal{S}||\mathcal{A}| \cdot \log \frac{H|\mathcal{S}||\mathcal{A}|}{\delta \cdot \epsilon} \cdot \log \frac{H}{\epsilon} \right]$$

*and obtains a policy $\pi$ such that, with probability at least $1 - \delta$,*

$$\forall h \in [H] : \boldsymbol{v}_h^* - \epsilon\mathbf{1} \le \boldsymbol{v}_h^\pi \le \boldsymbol{v}_h^*,$$

*where $\boldsymbol{v}_h^*$ is the optimal value of $\mathcal{M}$ at stage $h$. Moreover, the number of samples used by the algorithm is*

$$O\left[ \frac{1}{\epsilon^2} \cdot H^3 |\mathcal{S}||\mathcal{A}| \cdot \log \frac{H|\mathcal{S}||\mathcal{A}|}{\delta \cdot \epsilon} \cdot \log \frac{H}{\epsilon} \right].$$

### F.1 Sample Lower Bound On $H$-MDP

In this section we show that the sample complexity obtained by the algorithm in the last section is essentially tight. Our proof idea is simple, we will reduce the $H$-MDP problem to a discounted MDP problem. If there is an algorithm that solves an $H$-MDP to obtain an $\epsilon$-optimal value, it also gives an value function to the discounted MDP. Therefore, the lower bound of solving $H$-MDP inherits from that of the discounted MDP. The formal guarantee is presented in the following theorem.

**Theorem F.7.** *Let $\mathcal{S}$ and $\mathcal{A}$ be finite sets of states and actions. Let $H > 0$ be a positive integer and $\epsilon \in (0, 1/2)$ be an error parameter. Let $\mathcal{K}$ be an algorithm that, on input an $H$-MDP $\mathcal{M} \overset{\text{def}}{=} (\mathcal{S}, \mathcal{A}, P, \boldsymbol{r})$ with a sampling oracle, outputs a value function $\boldsymbol{v}_1$ for the first stage, such that $\|\boldsymbol{v}_1 - \boldsymbol{v}_1^*\|_\infty \le \epsilon$ with probability at least $0.9$. Then $\mathcal{K}$ calls the sampling oracle at least $\Omega(H^{-3}\epsilon^{-2}|\mathcal{S}||\mathcal{A}|/\log \epsilon^{-1})$ times on some input $P$ and $\boldsymbol{r} \in [0,1]^{\mathcal{S}}$.*

*Proof.* Let $s_0 \in \mathcal{S}$ be a state. Denote $\mathcal{S}' = \mathcal{S}\backslash\{s_0\}$ be a subset of $\mathcal{S}$. Let $\gamma \in (0,1)$ be such that $(1 - \gamma)^{-1}\log \epsilon^{-1} \le H$. Suppose we have an DMDP $\mathcal{M}' = (\mathcal{S}', \mathcal{A}, P', \gamma, \boldsymbol{r}')$ with a sampling oracle. Let $\boldsymbol{v}^{*'}$ be the optimal value function of $\mathcal{M}'$. Note that $\boldsymbol{v}^{*'} \in \mathbb{R}^{\mathcal{S}'}$. We will show, in the next paragraph, an $H$-MDP $\mathcal{M} = (\mathcal{S}, \mathcal{A}, P, H, \boldsymbol{r})$ with first stage value $\boldsymbol{v}_1^*$, such that $\|\boldsymbol{v}_1^*|_{\mathcal{S}'} - \boldsymbol{v}^{*'}\| \le \epsilon$. Therefore, an $\epsilon$-approximation of $\boldsymbol{v}_1^*$ gives a $2\epsilon$-approximation to $\boldsymbol{v}^*$. We show that $\mathcal{K}$ can be used to obtain an $\epsilon$-approximate value $\boldsymbol{v}_1$ for $\boldsymbol{v}_1^*$ of $\mathcal{M}$ and thus $\mathcal{K}$ inherits the lower bound for obtaining $(2\epsilon)$-approximated value for $\gamma$-DMDPs.

For $\mathcal{M}$, in each state $s \in \mathcal{S}'$, for any action there is a $(1 - \gamma)$ probability transiting to $s_0$ and $\gamma$ probability to do the original transitions in $\mathcal{M}'$; for $s_0$, no matter what action taken, it transits to

**Algorithm 3** FiniteHorizonRandomQVI

1: **Input:** $\mathcal{M} = (\mathcal{S}, \mathcal{A}, \boldsymbol{r}, \boldsymbol{P})$ with a sampling oracle, $\boldsymbol{v}_1^{(0)}, \boldsymbol{v}_2^{(0)}, \dots, \boldsymbol{v}_H^{(0)}, \pi^{(0)} : \mathcal{S} \times [H] \to \mathcal{A}, u, \delta \in (0, 1)$;

2: $\backslash\backslash u$ is the initial error, $\pi^{(0)}$ is the input policy, and $\delta$ is the error probability

3: **Output:** $\boldsymbol{v}_1, \boldsymbol{v}_2, \dots, \boldsymbol{v}_H, \pi$

4:

5: **INITIALIZATION:**

6: Let $m_1 \leftarrow c_1 H^3 u^{-2} \log(8|\mathcal{S}||\mathcal{A}|\delta^{-1})$ for constant $c_1$;

7: Let $m_2 \leftarrow c_2 H^2 \log[2H|\mathcal{S}||\mathcal{A}|\delta^{-1}]$ for constant $c_2$;

8: Let $\alpha_1 \leftarrow m_1^{-1} \log(8|\mathcal{S}||\mathcal{A}|\delta^{-1})$;

9: For each $(s, a) \in \mathcal{S} \times \mathcal{A}$, sample independent samples $s_{s,a}^{(1)}, s_{s,a}^{(2)}, \dots, s_{s,a}^{(m_1)}$ from $\boldsymbol{P}_{s,a}$;

10: Initialize $\boldsymbol{w}_h = \widetilde{\boldsymbol{w}}_h = \widehat{\boldsymbol{\sigma}}_h = \boldsymbol{Q}_h^{(0)} \leftarrow \mathbf{0}_{\mathcal{S} \times \mathcal{A}}$ for all $h \in [H]$, and $i \leftarrow 0$;

11: Denote $\boldsymbol{v}_{H+1} \leftarrow \mathbf{0}$ and $\boldsymbol{Q}_{H+1} \leftarrow \mathbf{0}$

12: **for** each $(s, a) \in \mathcal{S} \times \mathcal{A}, h \in [H]$ **do**

13: $\qquad \backslash\backslash$*Compute empirical estimates of* $\boldsymbol{P}_{s,a}^\top \boldsymbol{v}_h^{(0)}$ *and* $\boldsymbol{\sigma}_{\boldsymbol{v}_h^{(0)}}(s, a)$

14: $\qquad$ Let $\widetilde{\boldsymbol{w}}_h(s, a) \leftarrow \frac{1}{m_1} \sum_{j=1}^{m_1} \boldsymbol{v}_h^{(0)}(s_{s,a}^{(j)})$

15: $\qquad$ Let $\widehat{\boldsymbol{\sigma}}_h(s, a) \leftarrow \frac{1}{m_1} \sum_{j=1}^{m_1} (\boldsymbol{v}_h^{(0)})^2(s_{s,a}^{(j)}) - \widetilde{\boldsymbol{w}}_h^2(s, a)$

16:

17: $\qquad \backslash\backslash$*Shift the empirical estimate to have one-sided error*

18: $\qquad \boldsymbol{w}_h(s, a) \leftarrow \widetilde{\boldsymbol{w}}_h(s, a) - \sqrt{2\alpha_1 \widehat{\boldsymbol{\sigma}}_h(s, a)} - 4\alpha_1^{3/4} \|\boldsymbol{v}_h^{(0)}\|_\infty - (2/3)\alpha_1 \|\boldsymbol{v}_h^{(0)}\|_\infty$

19: Let $\boldsymbol{v}_{H+1} \leftarrow \mathbf{0}$ and $\boldsymbol{Q}_{H+1} \leftarrow \mathbf{0}$.

20:

21: **REPEAT:** $\backslash\backslash$*successively improve*

22: **for** $h = H, H - 1$ to $1$ **do**

23: $\qquad \backslash\backslash$*Compute* $\boldsymbol{P}_{s,a}^\top [\boldsymbol{v}_h - \boldsymbol{v}_h^{(0)}]$ *with one-sided error*

24: $\qquad$ Let $\widetilde{\boldsymbol{v}}_h \leftarrow \boldsymbol{v}_h \leftarrow \boldsymbol{v}(\boldsymbol{Q}_{h+1}), \widetilde{\pi}(\cdot, h) \leftarrow \pi(\cdot, h) \leftarrow \pi(\boldsymbol{Q}_{h+1}), \boldsymbol{v}_h \leftarrow \widetilde{\boldsymbol{v}}_h$;

25: $\qquad$ For each $s \in \mathcal{S}$, if $\widetilde{\boldsymbol{v}}_h(s) \leq \boldsymbol{v}_h^{(0)}(s)$, then $\boldsymbol{v}_h(s) \leftarrow \boldsymbol{v}_h^{(0)}(s)$ and $\pi(s, h) \leftarrow \pi^{(0)}(s, h)$;

26: $\qquad$ For each $(s, a) \in \mathcal{S} \times \mathcal{A}$, sample independent samples $\widetilde{s}_{s,a}^{(1)}, \widetilde{s}_{s,a}^{(2)}, \dots, \widetilde{s}_{s,a}^{(m_2)}$ from $\boldsymbol{P}_{s,a}$;

27: $\qquad$ Let $\boldsymbol{g}_h(s, a) \leftarrow m_2^{-1} \sum_{j=1}^{m_2} [\boldsymbol{v}_h(\widetilde{s}_{s,a}^{(j)}) - \boldsymbol{v}_h^{(0)}(\widetilde{s}_{s,a}^{(j)})] - H^{-1}u/8$;

28:

29: $\qquad \backslash\backslash$*Improve* $\boldsymbol{Q}_h$:

30: $\qquad \boldsymbol{Q}_h \leftarrow \boldsymbol{r} + \boldsymbol{w}_h + \boldsymbol{g}_h$;

31: **return** $\boldsymbol{v}_1, \boldsymbol{v}_2, \dots, \boldsymbol{v}_H, \pi$.

---

itself with probability 1. Formally, for each state $s, s' \in \mathcal{S}', a \in \mathcal{A}$, $P(s'|s, a) = \gamma \cdot P'(\cdot|s, a)$ and $P(s_0|s, a) = (1 - \gamma)$; $P(s'|s_0, a) = 0$ and $P(s_0|s_0, a) = 1$. For $\boldsymbol{r}$, we set $\boldsymbol{r}(s_0, \cdot) = \mathbf{0}$ and $\boldsymbol{r}(s, \cdot) = \boldsymbol{r}'(s, \cdot)$ for $s \in \mathcal{S}'$. It remains to show that $\|\boldsymbol{v}_1^*|_{\mathcal{S}'} - \boldsymbol{v}^*\|_\infty \leq \epsilon$. First we note that $\boldsymbol{v}(\boldsymbol{r}) = \boldsymbol{v}_H^* \leq \boldsymbol{v}^*$. Then, by monotonicity of the $\mathcal{T}$ operator, we have, for all $h \in [H - 1]$ and $s \in \mathcal{S}'$,

$$\boldsymbol{v}_h^*|_{\mathcal{S}'}(s) = \max_a [\boldsymbol{r}'(s, a) + \gamma \boldsymbol{P}_{s,a}^{'\top} \boldsymbol{v}_{h+1}^*] \leq \boldsymbol{v}^{*'}.$$

In particular, $\boldsymbol{v}_1^*|_{\mathcal{S}'} \leq \boldsymbol{v}^{*'}$. Since the optimal policy $\pi^{*'}$ of $\mathcal{M}'$ can be used as a policy for the $H$-MDP as a non-optimal one, we have

$$\boldsymbol{v}^* - \epsilon \cdot \mathbf{1} \leq \left[ 1 + \gamma \boldsymbol{P}_{\pi^{*'}} + \gamma^2 \boldsymbol{P}_{\pi^{*'}}^2 + \dots + \gamma^H \cdot \boldsymbol{P}_{\pi^{*'}}^H \right] \boldsymbol{r}^{\pi^{*'}} \leq \boldsymbol{v}_1^*|_{\mathcal{S}'}.$$

This completes the proof. $\qquad\qquad\qquad\qquad\qquad\qquad\qquad\qquad\qquad\qquad\qquad\qquad\square$

The above lower bound with our algorithm also implies a sample lower bound for an $\epsilon$-policy.

**Corollary F.8.** *Let $\mathcal{S}$ and $\mathcal{A}$ be finite sets of states and actions. Let $H > 0$ be a positive integer and $\epsilon \in (0, 1/2)$ be an error parameter. Let $\mathcal{K}$ be an algorithm that, on input an $H$-MDP*

$\mathcal{M} := (\mathcal{S}, \mathcal{A}, P, \boldsymbol{r})$ *with a sampling oracle, outputs a policy* $\pi : \mathcal{S} \times [H] \rightarrow \mathcal{A}$, *such that* $\forall h : \|\boldsymbol{v}_h^\pi - \boldsymbol{v}_h^*\|_\infty \leq \epsilon$ *with probability at least* $0.9$. *Then* $\mathcal{K}$ *calls the sampling oracle at least* $\Omega(H^{-3}\epsilon^{-2}|\mathcal{S}||\mathcal{A}|/\log \epsilon^{-1})$ *times on the worst case input* $P$ *and* $\boldsymbol{r} \in [0,1]^{\mathcal{S}}$.

## Footnotes

[6]$\widetilde{O}(f)$ denotes $O(f \cdot \log^{O(1)} f)$.