[Reviews · NeurIPS 2018]

Reviewer 1



In this paper the authors present PAC-RL results for finite-spaces discounted MDP with a generative sampling model (which can generate in O(1) time a state transition for each given state-action pair). They introduce a value-iteration type algorithm for finding an $\epsilon$-optimal policy, and prove that its running time and sample complexity are nearly the best possible and match, up to logarithmic factors, the lower bounds established previously in [AMK13]. These results are significant: They improve over a recent result in [SWWY18] by a factor of $(1 - \gamma)^{-1}$ ($\gamma$ is the discount factor), and they also fill a known gap in the earlier results of [AMK13] regarding the sample complexity of finding a near-optimal policy. Both the algorithm and its analysis are very interesting. They build upon the prior work [AMK13] and [SWWY18], and combine the algorithmic ideas and proof techniques from both papers in a highly non-trivial and original way. In addition, as an extension to these results, the authors also present a similar near-optimality result for finite-horizon MDP (the extension is given in an appendix). This is an exciting paper. The results are very strong, and the paper is very well-written. While the analysis is intricate and subtle, the authors have structured the paper well to give the reader the big picture, and they have succeeded in explaining the key ideas and proof steps very clearly. Detailed comments, mostly about minor presentation issues: 1. line 157-176, the last paragraph on p. 4: This discussion on the "Bernstein Technique" is a little confusing. The authors talk about the variance of the total return of a policy $\pi$ and the variance of $V^\pi(S_1)$, but the authors' algorithm is a value iteration algorithm and it does not evaluate any policy. The name "Bernstein Technique" seems to suggest that the technique is based on Bernstein's inequality. But the key inequality in this technique is the inequality in the authors' Lemma C.2 (Lemma 8 in [AMK13]), applied to the optimal policy. This inequality is a property of MDP, not a statistical property derived from the Bernstein inequality. I suggest the authors revise the last paragraph on p. 4 to explain more precisely the third technique in their work. 2. line 214-215, p.5: It would be better if the authors can say something about the nature of these constants c_1, c_2, c_3, especially c_2. 3. p. 7: In the for-loop of Algorithm 2, the subroutine HALFERR (Algorithm 1) also needs the values of $\delta$ and $u$ as inputs. The authors need to specify these values in Algorithm 2. 4. p. 480, p. 15: The two equations in this line are strange; the proof of the second inequality of Lemma C.3 doesn't seem valid. Of course, this error doesn't affect other results, since Lemma C.3 is the authors' alternative proof of Lemma C.2, which was already proved in [AMK13, Lemma 8]. On the other hand, the last step of the proof of [AMK13, Lemma 8] given in that reference skips some details and isn't obvious, so it will be good if the authors can fix Lemma C.3 and give the proof details of [AMK13, Lemma 8] (the authors' Lemma C.2). 5. p. 17: It seems in this proof of Lemma 4.4, the authors also need the argument in the proof of Lemma E.3. Also, in the statement of Lemma 4.4 on p. 6, should $1 - 2 \delta$ be $1 - R \delta$ instead? Other minor comments related to typos etc.: 1. Lemma 4.4, p. 216-219, p. 6: Should the definitions of $\pi^*$ and $P^\pi Q$ be given outside the lemma? Also, no need to repeat the definition of $Q^*$. 2. Table 2, p. 11: The first two rows are the same. 3. Before line 480, p. 15: It seems this part of the proof for the first inequality of Lemma C.3 can be simplified by applying Jensen's inequality to $(1 - \gamma)(I - \gamma P)^{-1} \sqrt{v}$.) 4. Lemma D.1, p. 15: Should the conclusion of the lemma include the words "with high probability"? 5. p. 19: The proof of Proposition E.3 is misplaced near the top of this page, before the proposition is stated. The proof of Theorem 4.6 is also misplaced and should be moved to Section E.2. 6. Some typos: -- $\gamma$ is missing in (3.1)-(3.3), p. 3-4. -- What is $\beta$ in line 224, p. 6? -- In the displayed equation just above line 507, p. 15, $n$ should be $m$. -- p. 17: Should $\epsilon$ in the proof be $u$ instead?

Reviewer 2



Summary: ------------- This paper considers sample complexity bounds for discounted MDPs given a generative model. An algorithm is provided for which an optimal sample complexity bound is proven. This bound is the same as the bound given by Azar, Munos & Kappen (2013), but also holds for epsilon bigger than 1/sqrt{S}, where S is the number of states. The algorithm has a run-time bound that equals the bound on the sample complexity. Assessment: ---------------- The paper combines techniques from the papers [AMK13] and [SWWY18] to close a small gap in the PAC bounds of [AMK13] for larger epsilon. This is basically done by an algorithm that given an input policy, after taking \tilde{O}(SA/(1-gamma)^3) samples returns a policy with half the error of the input policy. The paper is sufficiently well-written to be able to follow, although a few things could be made clearer (cf. comments below) and some proof-reading, streamlining and polishing would be needed. In particular, I found the discussion of the run-time complexity a bit confusing. I didn't have a closer look at the proofs in the Appendix, the main part at least contains the most important steps and the main ideas. Overall, this is an ok paper. It makes a bit of progress, even if it fills just a small (and in my opinion not that significant) gap in a research direction that I personally do not consider that important. Given that the submitted version is not quite ready for publication, in my view this is a borderline paper. Comments: --------------- - I'd skip the reference to the finite-horizon case from Abstract and refer in the Introduction directly to the Appendix, if there is not enough space to say anything about this setting in the main part of the paper. - The discussion of related work is sometimes a bit unclear. (I'd suggest to merge the respective parts of Sections 1 and 5. This would also give space to include more of the technical material or details about the work on the finite horizon case in the main part.) For example, in l.51 it is claimed that if one uses the methods of [AMK13] the sample complexity bound one obtains scales with (1-gamma)^{-5}. I did not understand this, as the bounds in the latter paper have a different dependence on gamma. In lines 284-294 it is said that the run-time complexity of [AMK13] "is dominated by the time needed to construct the empirical model". Does this refer to computing the estimated transition probabilities and rewards? That the run-time for epsilon = O(1/sqrt{S}) is "not sublinear in the size of the MDP model" (l.288) applies not only to [AMK13] but also to the bounds in the paper, right? - In l.4 of Alg.2, I guess v^(i-1) and pi^(i-1) are supposed to be the input parameters for the HALFERR routine. (I'd suggest to apply VR-QVI right away instead of introducing another algorithm HALFERR.) - The ALT 2012 paper of Hutter&Lattimore is missing in the references. - Are references [Ber13a] and [Ber13b] the same? Post-Rebuttal: ------------------ Thanks for the clarifications. Overall, I stick to my assessment for the moment. The used techniques are mainly present already in the references [AMK13] and [SWWY18], while the right gamma-dependence has already been shown in [AMK13]. I definitely think that the results should be published, I'm not so convinced that the paper has to be at NIPS.

Reviewer 3



This paper gives a new algorithm that can obtain the already known complexity lower bound whose dependency on (1-\gamma) is (1-\gamma)^{-3}. Previous work [SWWY18] can obtain (1-\gamma)^{-4} result by means of the monotonicity technique and the variance reduction technique. The Bernstein technique can remove the last factor (1-\gamma) by lowering the error accumulation. This is a very good work. However, there are a lot of typos, hindering the correct understanding of it. Especially, these typos make it very hard to check the proof. As a result, I think this paper is not ready to be published until its incorrectness can be corrected and modified. I list severe typos here. 1. What’s the purpose of your using \tilde{\pi}^{(i)} in line 25 of Algorithm 1? It seems that you don’t aim to update it. I think it is an intermediate quantity to update \pi^{(i)}(s) just like the way \tilde{v}^{(i)}(s) helping compute v^{(i)}(s). Additionally, what if the condition in line 26 in Algorithm 1 fails to be satisfied? From the proof of Lemma 4.4, I guess you will assign \tilde{v} to v and \tilde{\pi} to \pi, please complete it. 2. I wonder what \mu in the phrase“(1-\gamma)\mu”in line 162 means. Maybe it is a clerical error and should be u. The power of (1-\gamma) in line 226 should be -3. 3. The RHS of the first inequality of line 564 is wrong, since gamma will not appear in the denominator of the second term, if you plug line 562 into it. What’s more. In line 563, the constant upper bound for the first inequality is not 2 but one larger than 2, which is not harmful to the main result but less rigorous. The second equality of line 565 is incorrect, but it is fine to ignore this mistake and then focus on the inequality, which determines the order of m_1.